# PROMPT TUNING WITH DIFFUSION FOR FEW-SHOT PRE-TRAINED POLICY GENERALIZATION

## ABSTRACT

Offline Reinforcement Learning (RL) methods harness previous experiences to derive an optimal policy, forming the foundation for pre-trained large-scale models (PLMs). When encountering tasks not seen before, PLMs often utilize several expert trajectories as prompts to expedite their adaptation to new requirements. Though a range of prompt-tuning methods has been proposed to enhance the quality of prompts, these methods frequently face restrictions due to prompt initialization, which can significantly constrain the exploration domain and potentially lead to suboptimal solutions. To eliminate the reliance on the initial prompt, we shift our perspective towards the generative model, framing the prompt-tuning process as a form of conditional generative modeling, where prompts are generated from random noise. Our innovation, the Prompt Diffuser, leverages a conditional diffusion model to produce prompts of exceptional quality. Central to our framework is the approach to trajectory reconstruction and the meticulous integration of downstream task guidance during the training phase. Further experimental results underscore the potency of the Prompt Diffuser as a robust and effective tool for the prompt-tuning process, demonstrating strong performance in the meta-RL tasks.

## 1 INTRODUCTION

Over the last few years, PLMs have demonstrated remarkable efficacy across diverse domains, including high-resolution image generation from text descriptions (DALL-E (Ramesh et al., 2021), ImageGen (Saharia et al., 2022)) and language generation (GPT (Brown et al., 2020)). The success of PLMs in numerous applications has sparked interest in their potential application to decision-making tasks. Moreover, the advent of the prompt-tuning technique has further empowered PLMs to swiftly adapt to downstream tasks by fine-tuning only a small number of parameters across various model scales and tasks. This efficient and effective adaptation process has made prompt-tuning a promising approach for tailoring PLMs to specific decision-making scenarios.

In the realm of reinforcement learning (RL), offline decision-making assumes a critical role, facilitating the acquisition of optimal policies from trajectories gathered by behavior policies, all without requiring real-time interactions with the environment. Nonetheless, offline RL encounters formidable challenges concerning generalization to unseen tasks and the fulfillment of varying constraints, primarily due to distribution shifts (Mitchell et al., 2021). Recent research efforts, such as Gato (Reed et al., 2022) and other generalized agents (Lee et al., 2022), have explored the use of transformer-based architectures and sequence modeling techniques to address multi-task problems in offline RL. Utilizing prompt-tuning techniques, these methods can efficiently adapt to the target task by fine-tuning a relatively small number of parameters. Nevertheless, it's noteworthy that prompt-tuning methods often exhibit sensitivity to initialization (Hu et al., 2023b; Lester et al., 2021). When a random prompt is utilized for initialization, the PLM's exploration may become constrained within a limited region, leading the subsequent updating process to converge towards a sub-optimal prompt, as empirically demonstrated in Section 3.3. This sensitivity necessitates the pre-collection of expert trajectories, which in turn limits their applicability across a broader range of scenarios.

To eliminate the reliance on the initial prompt, we shift our perspective towards the generative model, framing the prompt-tuning process as a form of conditional generative modeling, where prompts are generated from random noise. This approach obviates the need to collect expert prompts, and the final quality of generated prompts is then determined by the parameters of the generative model, which can incorporate prior knowledge via pre-training on a fixed, pre-collected training dataset.

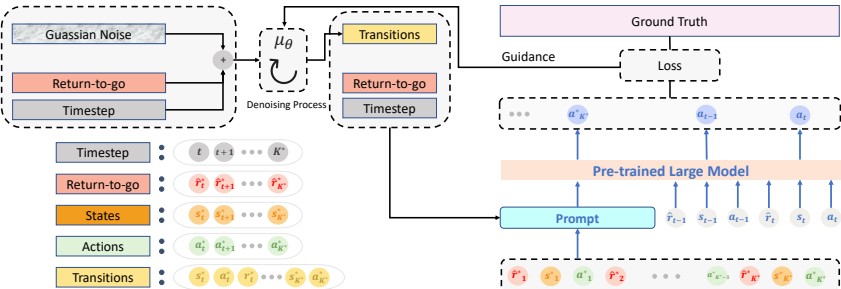

Figure 1: Overall architecture of Prompt Diffuser. Diffuser samples transitions while conditioning on the return-to-go and timestep tokens. These sampled transitions are then used to construct a prompt that is fed into the PLM. The loss between the predicted actions and the actual actions is employed to guide the denoising process, aiming to enhance the quality of the generated prompts.

However, in few-shot meta-learning environments, the quantity of offline data available from the target task is severely limited, necessitating rapid adaptability of the generative model to these tasks despite the small datasets. Moreover, the quality of these offline datasets typically falls short of expert quality, where the generative model must produce prompts that exceed the quality of the fine-tuning datasets, rather than simply generating prompts within the same distribution. Additionally, given the physical significance of trajectory prompts, even minor perturbations in the trajectory prompt can lead to substantial shifts in meaning (Hu et al., 2023b), highlighting the imperative need for precision in the generated prompts. All of these factors collectively contribute to the challenges encountered when applying this new paradigm to the prompt-tuning process.

To address these challenges, we introduce a novel algorithm named Prompt Diffuser (see Figure 1) which leverages a conditional diffusion model to produce prompts of exceptional quality. Within our framework for prompt generation, we establish the trajectory representation of Prompt Diffuser and adopt diffusion models to develop a generative model conditioned on returns, which ensures the precision of the generated prompts and expedites adaptability to new tasks (detailed in Section 4). Nevertheless, optimizing Prompt Diffuser solely with the DDPM loss can only achieve performance on par with the original dataset (Wang et al., 2022; Li et al., 2023). To augment the quality of prompt generation, we seamlessly incorporate guidance from downstream tasks into the reverse diffusion chain. By leveraging gradient projection techniques, the downstream task guidance is incorporated into the learning process without compromising the overall performance of the diffusion model, which is achieved by projecting the gradient of guidance loss onto the orthogonal direction to the subspace spanned by the diffusion loss. This novel approach successfully directs the generation of high-quality prompts, leading to improved performance in downstream tasks.

In summary, our work introduces a novel prompt-tuning framework that leverages diffusion models to generate high-quality prompts for RL agents. By incorporating downstream task guidance and employing gradient projection techniques, we successfully enhance the quality of generated prompts, leading to improved performance in downstream tasks. The experiments validate the effectiveness of our approach and demonstrate its potential for generating adaptive and transferable policies in meta-RL settings. Our contributions advance the field of prompt-tuning, providing a promising direction for optimizing pre-trained RL agents and improving their generalization and performance across various downstream tasks.

## 2 RELATED WORK

### 2.1 OFFLINE RL AS SEQUENCE MODELING

Offline RL has emerged as a prominent sequence modeling task and Transformer-based decision models have been applied to this domain. The primary objective is to predict the next sequence of actions by leveraging recent experiences, encompassing state-action-reward triplets. This approach can be effectively trained using supervised learning, making it highly suitable for offline RL and imitation learning scenarios. Various studies have investigated the utilization of Transformers in RL under the sequence modeling paradigm, including Gato (Reed et al., 2022), Multi-Game DT (Lee et al., 2022), Generalized DT (Furuta et al., 2021), Graph DT (Hu et al., 2023a), and the survey

work (Hu et al., 2022). In this paper, we propose a novel approach that builds upon the concepts of Prompt-DT (Xu et al., 2022; Hu et al., 2023b) while incorporating prompt-tuning techniques with diffusion models to significantly enhance the model's performance.

## 2.2 Diffusion Models in RL

Diffusion models (DMs) have emerged as a powerful family of deep generative models with applications spanning CV (Chen et al., 2023; Liu et al., 2023a), NLP (Li et al., 2022; Bao et al., 2022), and more recently, RL (Janner et al., 2022; Ajay et al., 2022). In particular, Diffuser (Janner et al., 2022) employs DM as a trajectory generator, effectively showcasing the potential of diffusion models in this domain. A consequent work (Ajay et al., 2022) introduces a more flexible approach by taking conditions as inputs to DM. This enhancement enables the generation of behaviors that satisfy diverse combinations of conditions, encompassing constraints and skills. Furthermore, Diffusion-QL (Wang et al., 2022) extends DM to precise policy regularization and injects the Q-learning guidance into the reverse diffusion chain to seek optimal actions. And Pearce et al. (2023); Reuss et al. (2023) apply DM in imitation learning to recover policies from the expert or human data without reward labels. Due to its capability to model trajectory distributions effectively, our study utilizes DM for generating prompts using few-shot trajectories, thereby ensuring the precision of generated prompts.

## 2.3 Prompt Learning

Prompt learning is a promising methodology in NLP involving the optimization of a small number of input parameters while keeping the main model architecture unchanged. The core concept of prompt learning revolves around presenting the model with a cloze test-style textual prompt, expecting the model to fill in the corresponding answer. Autoprompt (Shin et al., 2020) proposes an automatic prompt search methodology for efficiently finding optimal prompts, while recent advancements in prompt learning have incorporated trainable continuous embeddings for prompt representation (Li & Liang, 2021; Lester et al., 2021). Furthermore, prompt learning techniques have extended beyond NLP and the introduction of continuous prompts into pre-trained vision-language models has demonstrated significant improvements in few-shot visual recognition and generalization performance (Zhou et al., 2022b;a). In the context of RL, Prompt-tuning DT (Hu et al., 2023b) stands out for introducing prompt-tuning techniques using a gradient-free approach, aiming to retain environment-specific information and cater to specific preferences. In contrast, our approach leverages DM to directly generate prompts, incorporating flexible conditions during the generation process, which has resulted in notable performance improvements in the target tasks.

## 3 Preliminary

### 3.1 Prompt Decision Transformer

Transformer (Vaswani et al., 2017) has been increasingly investigated in RL using the sequence modeling pattern in recent years. Moreover, works from NLP suggest Transformers pre-trained on large-scale datasets demonstrate promising few-shot or zero-shot learning capabilities within the prompt-based framework (Liu et al., 2023b; Brown et al., 2020). Building upon this, Prompt-DT (Xu et al., 2022) extends the prompt-based framework to the offline RL setting, allowing for few-shot generalization to unseen tasks. In contrast to NLP, where text prompts can be transformed into conventional blank-filling formats to represent various tasks, Prompt-DT introduces the concept of trajectory prompts, leveraging few-shot demonstrations to provide guidance to the RL agent. A trajectory prompt comprises multiple tuples of state $s^*$, action $a^*$ and return-to-go $\hat{r}^*$, represented as $(s^*, a^*, \hat{r}^*)$, following the notation in (Chen et al., 2021). Each element marked with the superscript $\cdot^*$ is relevant to the trajectory prompt. Note that the length of the trajectory prompt is usually shorter than the task's horizon, encompassing only essential information to facilitate task identification, yet inadequate for complete task imitation. During training with offline collected data, Prompt-DT utilizes $\tau_i^{input} = (\tau_i^*, \tau_i)$ as input for each task $\mathcal{T}_i$. Here, $\tau_i^{input}$ consists of the $K^*$-step trajectory prompt $\tau_i^*$ and the most recent $K$-step history $\tau_i$, and is formulated as follows:

$$\tau_i^{input} = (\hat{r}_{i,1}^*, s_{i,1}^*, a_{i,1}^*, \ldots, \hat{r}_{i,K^*}^*, s_{i,K^*}^*, a_{i,K^*}^*, \ \hat{r}_{i,1}, s_{i,1}, a_{i,1}, \ldots, \hat{r}_{i,K}, s_{i,K}, a_{i,K}). \tag{1}$$

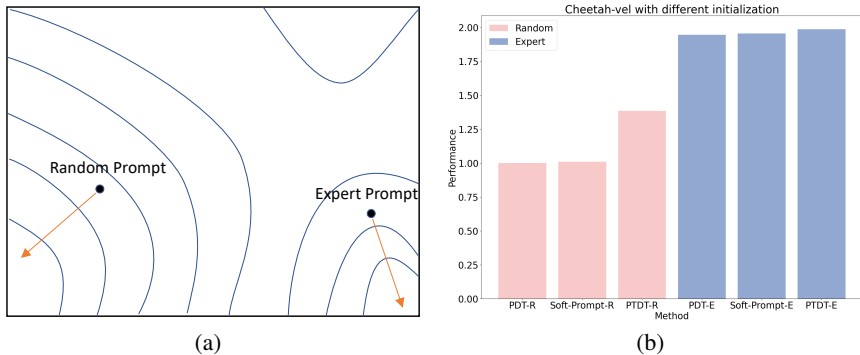

(a)                                         (b)

Figure 2: (a) The figure of the prompt updating process, where the prompt is treated as a point, and the updating process is simplified to identify the minimum value. (b) The performance of various methods under different prompt initialization conditions within the Cheetah-velocity environment.

The prediction head linked to a state token $s$ is designed to predict the corresponding action $a$. For continuous action spaces, the training objective aims to minimize the mean-squared loss:

$$L_{DT} = \mathbb{E}_{\tau_i^{input} \sim \mathcal{T}_i} \left[ \frac{1}{K} \sum_{m=1}^{K} (a_{i,m} - \pi(\tau_i^*, \tau_{i,m-1},))^2 \right]. \tag{2}$$

### 3.2 DIFFUSION MODELS

Diffusion models (Sohl-Dickstein et al., 2015; Ho et al., 2020) represent a particular class of generative models that learn the data distribution $q(x)$ from a dataset $\mathcal{D} := \{x_i\}$. The data-generation process is modeled through a two-stage process. In the forward diffusion chain, noise is gradually added to the data $x^0 \sim q(x)$ in $N$ steps, following a pre-defined variance schedule $\beta_i$, expressed as:

$$q(x^{1:N}|x^0) := \prod_{k=1}^{N} q(x^k|x^{k-1}), \ q(x^k|x^{k-1}) := \mathcal{N}(x^k; \sqrt{1-\beta_k}x^{k-1}, \beta_k \boldsymbol{I}). \tag{3}$$

The trainable reverse diffusion chain is formulated as:

$$p_\theta(x^{0:N}) := \mathcal{N}(x^N; \boldsymbol{0}, \boldsymbol{I}) \prod_{k=1}^{N} p_\theta(x^{k-1}|x^k), \ \ p_\theta(x^{k-1}|x^k) := \mathcal{N}(x^{k-1}; \mu_\theta(x^k, k), \Sigma_\theta(x^k, k)). \tag{4}$$

Ho et al. (2020) propose the simplified loss function in DDPMs for the diffusion timestep $k$:

$$L_k = \mathbb{E}_{k,x^0,\epsilon_k} \left[ ||\epsilon_k - \epsilon_\theta(\sqrt{\bar{\alpha}_k}x^0 + \sqrt{1-\bar{\alpha}_k}\epsilon_k, k)||^2 \right], \tag{5}$$

where $\bar{\alpha}_k = \prod_{i=1}^{k}(1-\beta_i)$, $\epsilon_\theta(\sqrt{\bar{\alpha}_k}x^0 + \sqrt{1-\bar{\alpha}_k}\epsilon_k, k)$ represents the noise predicted by the neural network and $\epsilon_k$ denotes the true noise utilized in the forward process. In this study, we harness the formidable capacity of diffusion models to reconstruct the data distribution, denoted as $q(x)$, for the generation of high-precision prompts.

### 3.3 RETHINKING THE RL PROMPTS

**What is the essence of RL prompts?** In NLP-based prompt learning, the fundamental assumption is that large language models have acquired sufficient knowledge from their pre-training data, and our task is to discover the most effective means of extracting this knowledge. However, in the realm of RL, it is impractical to assemble a corpus that comprehensively encompasses the diverse environments and tasks encountered. Thus RL agent is required to imitate the provided trajectory prompts, rather than using prompts to extract knowledge from the pre-trained model, which highlights the importance of curating high-quality prompts and is empirically detailed in Section 5.

**Why do we need to resort to generative models?** When a random prompt is utilized for initialization, the PLM's exploration may become constrained within a limited region, leading the subsequent

updating process to converge towards a sub-optimal prompt, which is illustrated in a simplified manner in Figure 2(a). In support of this assertion, we investigate various prompt-tuning methods with different prompt initializations in Cheetah-vel environments. As depicted in Figure 2(b), when the prompt is initialized with random quality, it is challenging to optimize it to expert quality, resulting in significantly poorer performance compared to expert initialization.

These observations advocate for a novel paradigm to replace the conventional prompt-tuning approach in the field of RL. Our solution is the generative model, which initially integrates prior knowledge through pre-training on training prompts datasets and directly generates prompts from random noise, thereby eliminating the dependence on initial prompts.

## 4 METHOD

Our objective is to maximize rewards in the target domain using PLM and conduct experiments on few-shot policy generalization tasks, evaluating the PLM's capability to generalize to new tasks. In line with suggestions from the fields of NLP and CV, fine-tuning the prompt for PLM proves to be more effective. However, the prompt-tuning approach remains relatively unexplored in the domain of RL, which presents new problems and challenges (Hu et al., 2023b). We formulate the prompt-tuning process as the standard problem of conditional generative modeling (GM):

$$\max_{\theta} \mathbb{E}_{s_0 \sim \rho_0} \left[ \sum_{t=1}^{T} r(s_t, \text{PLM}(s_{0:t}, a_{0:t-1}, \tau^*_{\text{prompt}})) \right],$$
$$\text{where} \quad \tau^*_{\text{prompt}} \sim \text{GM}_\theta(\tau^*_{\text{initial}} \mid C), \tag{6}$$

where condition $C$ could encompass various factors, such as the return achieved under the trajectory, the constraints met by the trajectory, or the skill demonstrated in the trajectory. Here we adopt the Prompt-DT as the PLM and MLP-based diffusion model as the GM. The trajectory is constructed based on the conditional diffusion process:

$$q(x^{k+1}(\tau^*) \mid x^k(\tau^*)), \quad p_\theta(x^{k-1}(\tau^*) | x^k(\tau^*), y(\tau^*)). \tag{7}$$

Here, $q$ denotes the forward noising process, while $p_\theta$ represents the reverse denoising process.

In the following sections, we first provide a detailed explanation of our approach employing a conditional diffusion model as an expressive GM for prompt generation. We then introduce diffusion loss, which acts as the behavior cloning term, constraining the distribution of the generated prompt to that of the training dataset. Lastly, we discuss the incorporation of downstream task guidance during the training aiming at enhancing the quality of the generated prompt. The overall pipeline of our Prompt Diffuser is depicted in Figure 1, providing a detailed presentation of the diffusion formulation, the corresponding diffusion loss, and the downstream task guidance.

### 4.1 DIFFUSION FORMULATION

In images, the diffusion process is applied across all pixel values in an image. Analogously, it may seem intuitive to apply a similar process to model the states and actions within a trajectory. To align with the input format Equation 1 required by the PLM, we formulate $x^0(\tau^*)$ as the transition sequence that encompasses states, actions, and rewards:

$$x^0(\tau^*) := \begin{bmatrix} s_t^* & s_{t+1}^* & \cdots & s_{t+H-1}^* \\ a_t^* & a_{t+1}^* & \cdots & a_{t+H-1}^* \\ r_t^* & r_{t+1}^* & \cdots & r_{t+H-1}^* \end{bmatrix}, \tag{8}$$

with the condition:

$$y(\tau^*) := \begin{bmatrix} \hat{r}_t^* & \hat{r}_{t+1}^* & \cdots & \hat{r}_{t+H-1}^* \\ t & t+1 & \cdots & t+H-1 \end{bmatrix}, \tag{9}$$

where $y(\tau^*)$ contains the returns-to-go $\hat{r}_t^* = \sum_{t'=t}^{T} r_{t'}^*$ and timesteps. Although the reward token $r_t^*$ is not directly utilized in the prompt formation 1, denoising the entire transition process can introduce model bias towards the inherent transition dynamics (He et al., 2023). Given the trajectory representation, one approach could involve training a classifier $p_\phi(y(\tau^*) \mid x^k(\tau^*))$ to predict $y(\tau^*)$ from noisy trajectories $x^k(\tau^*)$. However, it necessitates estimating a Q-function, which entails a separate, complex dynamic programming procedure. Instead, we opt to directly train a conditional

diffusion model conditioned on the $y(\tau^*)$ as per Equation 9. This conditioning empowers the model with the capacity to discern distinct tasks effectively, which expedites the pre-trained model's adaptation to novel tasks that exhibit analogous conditions.

## 4.2 Diffusion Loss

With the diffusion formulation, our objective is to generate a prompt trajectory that facilitates the rapid adaptation of the PLM to novel and unseen target tasks. By utilizing a collection of trajectory prompts derived from these unseen tasks, our intention is to extract sufficient task-specific information. This information is crucial for guiding the Prompt Diffuser in generating the most appropriate and effective prompt trajectory tailored to the intricacies of the target task. Thus during the training process, we adopt the approach from DDPM (Ho et al., 2020) and add extra conditions to train the reverse diffusion process $p_\theta$, which is parameterized by the noise model $\epsilon_\theta$ with loss:

$$L_{DM} = \mathbb{E}_{k \sim \mathcal{U}, \tau^* \sim \mathcal{D}, \epsilon_k \sim \mathcal{N}(\mathbf{0}, \mathbf{I})}[||\epsilon_k - \epsilon_\theta(\sqrt{\bar{\alpha}_k}x^0(\tau^*) + \sqrt{1 - \bar{\alpha}_k}\epsilon_k, y(\tau^*), k)||^2], \tag{10}$$

where $\mathcal{U}$ is a uniform distribution over the discrete set as $\{1, \ldots, N\}$ and $\mathcal{D}$ denotes the initial prompt dataset, collected by behavior policy $\pi_b$.

Next, we demonstrate that the loss term $L_{DM}$ functions as a behavior cloning term, effectively constraining the distribution of the generated prompt to match that of the training dataset. Suppose $x^0(\tau^*)$ is a continue random vector, and

$$\mathbb{P}(||x^0(\tau^*)||_2 \le R = N^{C_R}|\tau^* \sim \mathcal{D}) = 1, \tag{11}$$

for some arbitrarily large constant $C_R > 0$. We denote a total variation distance between two distributions $P_1$ and $P_2$ as $TV(P_1, P_2)$ and the upper bound of the total variation distance between the learned and true trajectory distributions is shown below (Li et al., 2023):

$$TV(q, p) \le \sqrt{\frac{1}{2}\text{KL}(q||p)} \le C_1 \frac{d^2 \log^3 N}{\sqrt{N}}. \tag{12}$$

As the diffusion timesteps $N$ increase, the generated prompt distribution $p$ progressively converges towards the training dataset $q$, thereby ensuring the precision of the generated prompts. However, it also imposes a limitation, as it prevents the generated prompts from surpassing the performance of the behavior trajectories contained within the offline dataset $\mathcal{D}$.

## 4.3 Diffusion Guidance

To improve the quality of the prompts, we incorporate downstream task guidance into the reverse diffusion chain during the training stage, with the objective of generating prompts that have a positive impact on the performance of the downstream tasks. With the output from the diffusion model, the loss for the downstream tasks can be formulated as follows:

$$L_{DT} = \mathbb{E}_{\tau_i^{input} \sim \mathcal{T}_i}\left[\frac{1}{K}\sum_{m=1}^{K}(a_{i,m} - \text{PLM}(x^0(\tau_i^*), y(\tau_i^*), \tau_{i,m}))^2\right]. \tag{13}$$

Note that $x^0(\tau^*)$ is sampled through Equation 4, allowing the gradient of the loss function with respect to the prompt to be backpropagated through the entire diffusion chain.

Nonetheless, a direct linear combination update of these two losses might lead to a deterioration in performance, potentially owing to the competitive interaction among distinct losses, which is elucidated in Section 5.4. Towards this end, we characterize the correlation between the two loss subspaces, employing gradient projection as the analytical tool. Specifically, let $S_{DM}^\perp = \text{span}\{B\} = \text{span}\{[u_1, \ldots, u_M]\}$ represent the subspace spanned by $\nabla L_{DM}^\perp$, where $B$ constitutes the bases for $S_{DM}^\perp$ and $(\cdot)^\perp$ denotes the orthogonal space (consisting of a total of $M$ bases extracted from $\nabla L_{DM}^\perp$). For any matrix $A$ with a suitable dimension, denote its projection onto subspace $S_{DM}^\perp$ as:

$$\text{Proj}_{S_{DM}^\perp}(A) = ABB^\top, \tag{14}$$

where $(\cdot)^\top$ is the matrix transpose. Based on the Equation 14, the final update gradient can be shown

$$\nabla L = \nabla L_{DM} + \text{Proj}_{S_{DM}^\perp}(\nabla L_{DT}) \tag{15}$$

Note that $L_{DM}$ can be efficiently optimized by sampling a single diffusion step $i$ for each data point, but $L_{DT}$ necessitates iteratively computing $\epsilon_\theta$ networks $N$ times, which can potentially become a bottleneck for the running time and may lead to vanishing gradients. Thus we restrict the value of $N$ to a relatively small value. We also provide the theoretic support for our gradient projection technique in Appendix G.

After the training process, we adopt the low-temperature sampling technique (Ajay et al., 2022) to produce high-likelihood sequences:

$$x^{k-1}(\tau^*) \sim \mathcal{N}(\mu_\theta(x^k(\tau^*), y(\tau^*), k), \beta\Sigma_k) \tag{16}$$

where the variance is reduced by $\beta \in [0,1)$ for generating better quality sequences and $\mu_\theta(x^k(\tau^*), y(\tau^*), k)$ is constructed as:

$$\mu_\theta(x^k(\tau^*), y(\tau^*), k) = \frac{1}{\sqrt{\alpha_k}}(x^k(\tau^*) - \frac{\beta_k}{\sqrt{1-\bar{\alpha}_k}}\epsilon_\theta(x^k(\tau^*), y(\tau^*), k)) \tag{17}$$

Overall, pseudocode for the conditional prompt diffuser method is given in Algorithm 1.

---

**Algorithm 1** Prompt Diffuser

---

**Require:** Initial prompt $\tau^*_{\text{initial}}$, diffuser network $\mu_\theta$, sample timesteps $N$
1: // train the prompt diffuser
2: **for** each iteration **do**
3:     Construct the $x^0(\tau^*_{\text{initial}})$ and $y(\tau^*_{\text{initial}})$ by Equation 8 and Equation 9.
4:     Compute the diffusion model loss $L_{DM}$ by Equation 10.
5:     Compute the downstream task loss $L_{DT}$ by Equation 13.
6:     Update the diffuser network $\mu_\theta$ by Equation 15 if $\nabla L_{DM} \cdot \nabla L_{DT} < 0$ else $\nabla L = \nabla L_{DM} + \nabla L_{DT}$.
7: **end for**
8: // inference with prompt diffuser
9: initialize prompt $x^N(\tau^*) \sim \mathcal{N}(\mathbf{0}, \boldsymbol{I})$ and $y(\tau^*) \leftarrow y(\tau^*_{\text{initial}})$
10: **for** $k = N, \ldots, 1$ **do**
11:     $\mu \leftarrow \mu_\theta(x^k(\tau^*), y(\tau^*), k)$
12:     Sample $x^{k-1}(\tau^*)$ with $\mu$ by Equation 16
13: **end for**
14: Output the prompt $x^0(\tau^*)$ and $y(\tau^*)$ for downstream tasks.

---

## 5 EXPERIMENT

We perform extensive experiments to assess the ability of Prompt Diffuser by using the episode accumulated reward as the evaluation metric. Our experimental evaluation seeks to answer the following research questions: (1) Does Prompt Diffuser improve the model generalization by generating a better prompt? (2) How does the quality of the prompt datasets influence the effectiveness of the Prompt Diffuser? (3) Does the diffusion guidance successfully facilitate the downstream task performance without disrupting the DDPM update progress?

### 5.1 ENVIRONMENTS AND OFFLINE DATASETS

To ensure a fair comparison with Prompt-Tuning DT (Hu et al., 2023b), we select four distinct meta-RL control tasks: Cheetah-dir, Cheetah-vel, Ant-dir, and Meta-World reach-v2. In the Cheetah-dir and Ant-dir tasks, the objective is to incentivize high velocity in the goal direction. On the other hand, the Cheetah-vel task penalizes deviations from the target velocity using l2 errors. The Meta-World benchmark (Yu et al., 2020) includes table-top manipulation tasks that require a Sawyer robot to interact with various objects. For our evaluation, we utilized the Meta-World reach-v2 benchmark, which comprises 45 training tasks for pre-training the Prompt-DT and the Prompt Diffuser. Subsequently, the testing set, consisting of 5 tasks with different goal positions, is employed for further fine-tuning the Prompt Diffuser. We followed the dataset construction and settings outlined in (Hu et al., 2023b) for the meta-RL control tasks considered in this study.

### 5.2 BASELINES

We compare our proposed Prompt Diffuser with six baseline methods to address the aforementioned questions. For each method, we assess task performance based on the episode accumulated reward

Table 1: Results for meta-RL control tasks. The best mean scores are highlighted in bold (within few-shot settings). For each environment, prompts of length $K^* = 5$ are utilized, and fine-tuning is conducted using few-shot expert data collected from the target task with three different seeds. Notably, Prompt Diffuser achieves the best average result during all few-shot fine-tuning methods.

| | Prompt-DT-Random | Prompt-DT-Expert | MT-ORL | Soft Prompt | Adaptor | Prompt-DT-FT | Prompt-Tuning DT | Prompt Diffuser | Prompt-DT-FT-Full |
|---|---|---|---|---|---|---|---|---|---|
| Sizes | - | - | - | 0.24K | 13.49K | 13.71M | 0.24K | 0.17M | 13.71M |
| Percentage | - | - | - | 0.0018% | 0.098% | 100% | 0.0018% | 1.24% | 100% |
| Cheetah-dir | $935.3 \pm 2.6$ | $934.6 \pm 4.4$ | $-46.2 \pm 3.4$ | $940.2 \pm 1.0$ | $875.2 \pm 4.2$ | $936.9 \pm 4.8$ | $941.5 \pm 3.2$ | $\mathbf{945.3 \pm 7.2}$ | $950.6 \pm 1.2$ |
| Cheetah-vel | $-127.7 \pm 9.9$ | $-43.5 \pm 3.4$ | $-146.6 \pm 2.1$ | $-41.8 \pm 2.1$ | $-63.8 \pm 6.3$ | $-40.1 \pm 3.8$ | $-39.5 \pm 3.7$ | $\mathbf{-35.3 \pm 2.4}$ | $-23.0 \pm 1.1$ |
| Ant-dir | $278.7 \pm 38.7$ | $420.2 \pm 5.1$ | $110.5 \pm 2.2$ | $379.1 \pm 33.7$ | $361.4 \pm 5.6$ | $425.2 \pm 8.6$ | $427.9 \pm 4.3$ | $\mathbf{432.1 \pm 6.7}$ | $494.2 \pm 2.3$ |
| MW reach-v2 | $457.2 \pm 30.3$ | $469.8 \pm 29.9$ | $264.1 \pm 9.6$ | $448.7 \pm 41.3$ | $477.9 \pm 2.1$ | $478.1 \pm 27.8$ | $472.5 \pm 29.0$ | $\mathbf{555.7 \pm 6.8}$ | $564.7 \pm 0.7$ |
| **Average** | 385.9 | 445.3 | 45.5 | 431.6 | 412.7 | 450.0 | 450.3 | **474.4** | 496.6 |

Table 2: Ablation on the effect of prompt initialization on the prompt-tuning methods. In the Cheetah-vel environment, we vary the quality of both prompts and datasets across three levels: Expert, Medium, and Random. While conventional prompt-tuning approaches are influenced by prompt initialization, our method overcomes this limitation by treating it as a generative problem.

| Prompt Initialization | Prompt-Tuning DT | | | Prompt Diffuser | | |
|---|---|---|---|---|---|---|
| | Exp. Prompt | Med. Prompt | Ran. Prompt | Exp. Prompt | Med. Prompt | Ran. Prompt |
| Exp. Dataset | $-42.1 \pm 1.3$ | $-49.9 \pm 1.6$ | $-89.2 \pm 5.8$ | $-32.6 \pm 1.3$ | $-34.2 \pm 2.8$ | $-33.5 \pm 2.0$ |
| Med. Dataset | $-41.4 \pm 0.8$ | $-50.3 \pm 1.9$ | $-90.3 \pm 5.3$ | $-34.2 \pm 2.8$ | $-33.5 \pm 2.0$ | $-34.4 \pm 2.4$ |
| Ran. Dataset | $-41.2 \pm 0.8$ | $-50.4 \pm 1.9$ | $-90.4 \pm 4.8$ | $-33.5 \pm 2.0$ | $-33.2 \pm 2.2$ | $-32.5 \pm 1.6$ |

in each testing task. To ensure a fair comparison, all fine-tuning methods utilize the same PLM. The baseline methods are as follows (detailed in Appendix F): (1) **Prompt-DT** exclusively employs the trajectory prompt for the target task without any additional fine-tuning process during testing. Our evaluation includes distinct experiments employing random and expert prompts. (2) **MT-ORL** omits the prompt augmentation step used in Prompt-DT to construct a variant of the approach. (3) **Soft Prompt** treats prompt as a "soft prompt" and updates it using the AdamW optimizer, analogous to a common practice in the NLP domain. (4) **Adaptor** plugs an adaptor module into each decoder layer, inspired by HDT (Xu et al., 2023), except for the hyper-network used for initialization. (5) **Prompt-Tuning DT** (Hu et al., 2023b) represents the first application that incorporates prompt tuning techniques in the RL domain, catering to specific preferences in the target environment with preference ranking. (6) **Prompt-DT-FT** fine-tunes the entire model parameters of the pre-trained model during testing, utilizing a limited amount of data from the target task. The performance of the **full-data settings** is also presented, serving as an upper bound for all fine-tuning methods.

## 5.3 MAIN RESULTS

We conduct a comparative analysis between the Prompt Diffuser and the baseline methods to evaluate their few-shot generalization capabilities and assess the tuning efficiency of the Prompt Diffuser in comparison to the other fine-tuning approach. For evaluation, we employ the averaged episode accumulated reward in the test task set as the metric. The primary results are summarized in Table 1, which presents the few-shot performance of different algorithms.

The outcomes of MT-ORL underscore its inability to attain high rewards in the target tasks, thus highlighting the crucial role of prompt assistance in facilitating the PLM's adaptation to these specific tasks. The comparison between random and expert prompts in Prompt-DT further accentuates the necessity of high-quality prompts, which serves as a strong rationale for the development of our prompt-tuning techniques. Among the parameter-efficient fine-tuning methods, only Prompt-Tuning DT manages to achieve performance comparable to Prompt-DT-FT. Importantly, our proposed approach exhibits significant performance improvements over both methods, even nearing the upper bound established by fine-tuning conducted on the full-data settings. This serves as a vivid demonstration of the distinct advantages offered by our innovative prompt-tuning techniques.

## 5.4 ABLATION

We perform several ablation studies to examine particular aspects of our Prompt Diffuser, with a primary focus on the meta-RL control environments. These ablation studies are designed to offer

insights into essential factors related to common prompt-tuning methods, such as prompt initialization. This analysis enables us to highlight the advantages and drawbacks of our proposed approach in comparison to other prompt-tuning techniques. More ablation study about our Prompt Diffuser and the visualization could be found in Appendix H and I.

**Prompt Initialization.** Prompt initialization plays a crucial role in guiding the agent's behavior and shaping the learning process (Gu et al., 2021). To investigate its impact, we conduct an ablation study in the Cheetah-vel environment. The results of this ablation study are presented in Table 2. Compared to Prompt-Tuning DT, which exhibits sensitivity to prompt quality, the Prompt Diffuser method displays robustness to variations in training data and prompt initialization. The key factor contributing to this robustness lies in the fundamental difference in the optimization strategies employed by the two methods. In the case of Prompt-Tuning DT, the prompt is directly updated during the optimization process. On the other hand, the Prompt Diffuser adopts a different approach by modeling the prompt-tuning process as conditional generative models, which are trained on the prompt with the additional guidance of downstream task loss. This distinctive formulation allows the Prompt Diffuser to refine and improve the prompt quality through the conditioning of downstream task loss, even when fine-tuning with relatively poor random prompt distributions. This finding provides valuable insights into the potential of Prompt Diffuser for prompt-tuning tasks, as it demonstrates the ability to achieve enhanced performance without relying solely on an expert dataset or a highly optimized initial prompt.

**Diffusion Guidance.** To improve the quality of the prompts, we propose a novel approach that involves integrating downstream task guidance into the reverse diffusion chain, while carefully preserving the continuity of the DDPM update progress. Leveraging the gradient projection technique, we successfully integrate the downstream task information into the learning process without compromising the overall performance of the Diffusion model. To demonstrate the effectiveness of our proposed method, we conduct three additional variants with different update gradients: Equation 18 solely trains using the $L_{DM}$ loss without incorporating downstream guidance, Equation 19 solely trains using the $L_{DT}$ to investigate the effect of guidance, and Equation 20 combines both the downstream task guidance loss and the $L_{DM}$ loss gradients in a simple manner:

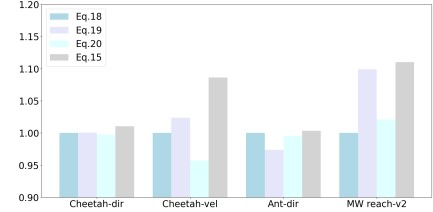

$$\nabla L = \nabla L_{DM} \tag{18}$$

$$\nabla L = \nabla L_{DT} \tag{19}$$

$$\nabla L = \nabla L_{DM} + \nabla L_{DT} \tag{20}$$

Figure 3: Ablation on diffusion guidance. We establish the performance of Equation 18 as the baseline and subsequently present the relative performance. Remarkably, our gradient projection technique consistently yields the most favorable outcomes.

As illustrated by the outcomes depicted in Figure 3, the guidance loss assumes a pivotal role, particularly evident in the MW reach-v2 environment, where it yields robust performance. Furthermore, the direct addition of the loss through Equation 20 results in inferior outcomes compared to exclusively employing DM loss and DT loss. In contrast, our proposed method adeptly circumvents potential conflicts between losses, thereby contributing to improved results in all environments.

## 6 CONCLUSION

We introduce Prompt Diffuser, a methodology that leverages a conditional diffusion model to generate superior-quality prompts. This approach shifts the traditional prompt-tuning process towards a conditional generative model, where prompts are generated from random noise. Through our trajectory reconstruction model and gradient projection techniques, Prompt Diffuser effectively overcomes the need for pre-collecting expert prompts and facilitates the PLM's efficient adaptation to novel tasks using generated prompts. Extensive experimental results demonstrate the substantial performance advantage of our approach over other parameter-efficient methods, approaching the upper bound in performance. We anticipate that our work will pave the way for the application of prompt-tuning techniques in the realm of RL, offering a generative model perspective.

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

# Appendices

## A  DETAILED ENVIRONMENT

We evaluate our approach on a variety of tasks, including meta-RL control tasks. These tasks can be described as follows:

- Cheetah-dir: The task comprises two directions: forward and backward, in which the cheetah agent is incentivized to attain high velocity along the designated direction. Both the training and testing sets encompass these two tasks, providing comprehensive coverage of the agent's performance.
- Cheetah-vel: In this task, a total of 40 distinct tasks are defined, each characterized by a different goal velocity. The target velocities are uniformly sampled from the range of 0 to 3. The agent is subjected to a penalty based on the $l_2$ error between its achieved velocity and the target velocity. We reserve 5 tasks for testing purposes and allocate the remaining 35 tasks for training.
- Ant-dir: There are 50 tasks in Ant-dir, where the goal directions are uniformly sampled in a 2D space. The 8-joint ant agent is rewarded for achieving high velocity along the specified goal direction. We select 5 tasks for testing and use the remaining tasks for training.
- Meta-World reach-v2: This task involves controlling a Sawyer robot's end-effector to reach a target position in 3D space. The agent directly controls the XYZ location of the end-effector, and each task has a different goal position. We train on 15 tasks and test on 5 tasks.

By evaluating our approach on these diverse tasks, we can assess its performance and generalization capabilities across different control scenarios.

The generalization capability of our approach is evaluated by examining the task index of the training and testing sets, as shown in Table 3. The experimental setup in Section 5 adheres to the training and testing division specified in Table 3. This ensures consistency and allows for a comprehensive assessment of the approach's performance across different tasks.

Table 3: Training and testing task indexes when testing the generalization ability in meta-RL tasks.

| Cheetah-dir | |
|---|---|
| Training set of size 2 | [0,1] |
| Testing set of size 2 | [0.1] |
| **Cheetah-vel** | |
| Training set of size 35 | [0-1,3-6,8-14,16-22,24-25,27-39] |
| Testing set of size 5 | [2,7,15,23,26] |
| **ant-dir** | |
| Training set of size 45 | [0-5,7-16,18-22,24-29,31-40,42-49] |
| Testing set of size 5 | [6,17,23,30,41] |
| **Meta-World reach-v2** | |
| Training set of size 15 | [1-5,7,8,10-14,17-19] |
| Testing set of size 5 | [6,9,15,16,20] |

## B  IMPLEMENTATION DETAILS

We construct our policy as an MLP-based conditional diffusion model. Following the parameterization approach of Nichol & Dhariwal (2021), we design $\epsilon_\theta$ as a 3-layer MLP with Mish activations, utilizing 256 hidden units for all network layers. The input to $\epsilon_\theta$ consists of concatenated states, actions, and rewards from the prompt, with the return-to-go and timesteps serving as conditions. It is important to note that different tokens possess varying ranges, necessitating their normalization

to fall within the [-1, 1] interval, based on the respective tokens' maximum and minimum values. During the training phase, we employ prompts from training tasks to pre-train the diffusion model. Subsequently, we fine-tune the diffusion model using prompts from test tasks, a strategy that effectively accelerates the fine-tuning process.

## C  HYPERPARAMETERS

In the case of the Prompt Diffuser, our primary focus centers around the hyperparameters associated with the improved-diffusion (`https://github.com/openai/improved-diffusion`), with comprehensive details provided in Table 4. These hyperparameters have been thoughtfully selected to ensure the optimal performance of our proposed approach. The careful consideration and tuning of these hyperparameters contribute to the effectiveness and robustness of the Prompt Diffuser across various tasks and scenarios.

Table 4: Hyperparameters of Prompt Diffuser.

| Hyperparameter | Value |
| --- | --- |
| Diffusion steps | 100 |
| Learn sigma | False |
| Small sigma | True |
| Sample scheduler | Uniform |
| Loss function | MSE |
| Prompt length $K^*$ | 5 |
| Return-to-go conditioning | 1500 Cheetah-dir |
| | 0 Cheetah-vel |
| | 500 Ant-dir |
| | 650 MW reach-v2 |
| Fine-tune epochs | 20 |
| Fine-tune steps | 100 |
| Fine-tune batches | 32 |
| Dropout | 0.1 |
| Learning rate | $1 \times 10^{-4}$ |
| Adam betas | $(0.9, 0.95)$ |
| Grad norm clip | 0.25 |
| Weight decay | $1 * 10^{-4}$ |

## D  ABLATION ON THE GENERATIVE MODELS

Furthermore, we conduct additional experiments wherein we systematically explore the generative model's capabilities. The corresponding results are presented in the Table 5. It is noteworthy that while the model generating the prompts is varied, we maintain consistency across other settings, including the PLMs, hyperparameters, and the process of updating losses. This meticulous approach ensures a fair and unbiased comparison.

As the results demonstrate, our approach's superiority can be attributed to the efficacy of the diffusion model. Given that both methods consider downstream task losses, the observed distinctions between their performances can likely be attributed to inherent characteristics of the generative model itself. This model's capacity to generate in-distribution outcomes, closely resembling those in fine-tuned datasets, is of paramount importance. The sensitivity of prompt perturbations to final outcomes underscores the significance of precision in prompt generation.

While the transformation of prompt-tuning into a canonical conditional generative modeling problem alleviates the need for meticulous pre-collection of high-quality prompts, it places a considerable burden on the generative model itself. This model must meet elevated precision standards for prompt generation. However, through the strategic utilization of the highly expressive diffusion model, our approach exhibits substantial potential within the prompt-tuning domain. This poten-

Table 5: Comparison among various generative models. Each environment is fine-tuned using the limited trajectory samples with three random seeds.

| | Cheetah-dir | Cheetah-vel | Ant-dir | MW reach-v2 | Average |
|---|---|---|---|---|---|
| DM | $945.3 \pm 7.2$ | $-35.3 \pm 2.4$ | $432.1 \pm 6.7$ | $555.7 \pm 6.8$ | 474.4 |
| VAE | $927.2 \pm 6.5$ | $-47.3 \pm 2.1$ | $383.5 \pm 9.8$ | $511.5 \pm 8.8$ | 443.7 |

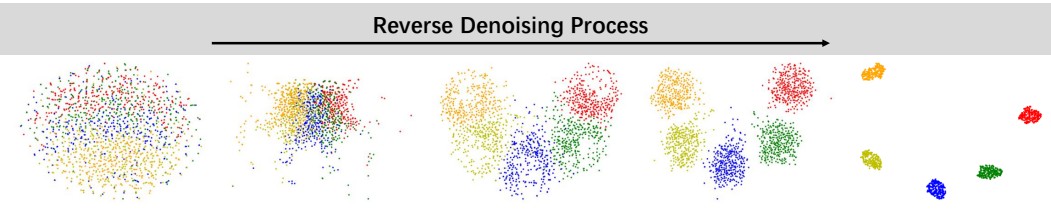

Figure 4: The visual results of the reverse denoising process. The initial samples $x^0(\tau)$ originate from Gaussian noise, with a prompt length of 5, represented by distinct colors.

tial is evidenced by our method's remarkable performance, outperforming other parameter-efficient approaches by a significant margin, as depicted in Table 1.

## E  MORE VISUAL RESULTS

We extend our insights into the reverse denoising process through Figure 4. As the denoising steps increase, the differentiation between individual timesteps becomes progressively clearer. This dynamic visualization serves to offer a deeper comprehension of the denoising process, highlighting how each step contributes to the refinement of the generated prompts.

## F  THE DETAILS OF BASELINES

In this section, we describe the implementation details of the baselines:

- **Prompt-DT**. Prompt-DT (Xu et al., 2022) is the pioneering application of sequence-prediction models that achieve state-of-the-art offline meta-RL algorithms. During testing, it exclusively employs the trajectory prompt for the target task without any additional fine-tuning process. Our evaluation includes distinct experiments employing random and expert prompts. We borrow the code from `https://github.com/mxu34/prompt-dt` for implementation.

- **MT-ORL**. Multi-task Offline RL omits the prompt augmentation step used in Prompt-DT to construct a variant of the approach. We keep other hyperparameters and implementation details the same as the official version except for the prompt augmentation.

- **Soft Prompt**. We consider the prompt as a "soft prompt" and update it using the AdamW optimizer, analogous to a common practice in the NLP domain. The learning rate and weight decay are set to $1 \times 10^{-4}$.

- **Adaptor**. An adaptor module is introduced to each decoder layer, inspired by HDT (Xu et al., 2023), except for the hyper-network used for initialization. The adaptor contains a down-projection layer $D_l$, a GELU nonlinearity, an up-projection layer $U_l$, and a feature-wise linear modulation (FiLM) layer $FiLM_l$ (Perez et al., 2018). The hidden dimension is set to 16.

- **Prompt-Tuning DT**. Prompt-Tuning DT (Hu et al., 2023b) represents the first application that incorporates prompt tuning techniques in the RL domain, catering to specific preferences in the target environment with preference ranking. We maintain consistency with the hyperparameters outlined in Hu et al. (2023b) and implement the offline settings algorithm.

- **Prompt-DT-FT**. We fine-tune the entire model parameters of the pre-trained Prompt-DT during testing, utilizing a limited amount of data from the target task. The performance of the full-data settings is also presented, serving as an upper bound for all fine-tuning methods. The fine-tuning

optimizer utilizes AdamW with a learning rate and weight decay of $1 \times 10^{-4}$. In the context of limited data availability, we conduct fine-tuning with 20 epochs in the few-data setting. This cautious approach allows us to make the most of the available data while ensuring a reasonable adaptation process. In contrast, in the full-data setting, the fine-tuning process is extended to 100 epochs to facilitate comprehensive adaptation with a larger dataset.

# G  THEORETIC SUPPORT

In this section, we give the following theoretical support for our gradient projection technique for the final performance.

We use $\mathcal{L}_1$ and $\mathcal{L}_2$ to denote $L_{DM}$ and $L_{DT}$ respectively for simplicity.

**Definition 1.** *We define $\phi_{ij}$ as the angle between two task gradients $\mathbf{g}_i$ and $\mathbf{g}_j$. We define the gradients as **conflicting** when $\cos \phi_{ij} < 0$.*

**Definition 2.** *Consider two task loss functions $\mathcal{L}_1 : \mathbb{R}^n \to \mathbb{R}$ and $\mathcal{L}_2 : \mathbb{R}^n \to \mathbb{R}$. We define the two-task learning objective as $\mathcal{L}(\theta) = \mathcal{L}_1(\theta) + \mathcal{L}_2(\theta)$ for all $\theta \in \mathbb{R}^n$, where $\mathbf{g_1} = \nabla \mathcal{L}_1(\theta)$, $\mathbf{g_2} = \nabla \mathcal{L}_2(\theta)$, and $\mathbf{g} = \mathbf{g_1} + \mathbf{g_2}$.*

Motivated by the work of Yu et al. (2020) on gradient-based techniques, we present Theorem 1 in this study. The convergence to a point where $\cos(\phi) = -1$ indicates a scenario where the gradients of $L_{DM}$ and $L_{DT}$ are in opposite directions. This situation typically arises when there is a conflict between the objectives of the two loss functions. In practical implementation, our priority is to ensure that the DM loss converges normally, even if it means sacrificing the convergence of the DT loss to some extent. This is because the $L_{DM}$ loss plays a crucial role in approximating the distribution of the existing dataset, which sets a foundational benchmark for the quality of the generated prompts. On the other hand, convergence to $L(\theta^*)$ signifies that the optimization process successfully finds the optimal point that minimizes the combined loss. This outcome is the ideal scenario, demonstrating the effectiveness of our gradient projection technique in jointly optimizing both loss functions.

This theoretical framework addresses a critical challenge in combining multiple loss functions, particularly when these functions have potentially conflicting gradients. Our approach ensures stable and effective optimization, even in complex scenarios where balancing multiple objectives is necessary. To enhance readability, we present the full proof for reference purposes.

**Theorem 1.** *Assume $\mathcal{L}_1$ and $\mathcal{L}_2$ are convex and differentiable. Suppose the gradient of $\mathcal{L}$ is $L$-Lipschitz with $L > 0$. Then, the gradient projection technique with step size $t \leq \frac{1}{L}$ will converge to either (1) a location in the optimization landscape where $\cos(\phi_{12}) = -1$ or (2) the optimal value $\mathcal{L}(\theta^*)$.*

*Proof.* We will use the shorthand $|| \cdot ||$ to denote the $L_2$-norm and $\nabla \mathcal{L} = \nabla_\theta \mathcal{L}$, where $\theta$ is the parameter vector. let $\mathbf{g_1} = \nabla \mathcal{L}_1$, $\mathbf{g_2} = \nabla \mathcal{L}_2$, $\mathbf{g} = \nabla \mathcal{L} = \mathbf{g_1} + \mathbf{g_2}$, and $\phi_{12}$ be the angle between $\mathbf{g_1}$ and $\mathbf{g_2}$.

Our assumption that $\nabla \mathcal{L}$ is Lipschitz continuous with constant $L$ implies that $\nabla^2 \mathcal{L}(\theta) - LI$ is a negative semi-definite matrix. Using this fact, we can perform a quadratic expansion of $\mathcal{L}$ around $\mathcal{L}(\theta)$ and obtain the following inequality:

$$\mathcal{L}(\theta^+) \leq \mathcal{L}(\theta) + \nabla \mathcal{L}(\theta)^T(\theta^+ - \theta) + \frac{1}{2}\nabla^2 \mathcal{L}(\theta)||\theta^+ - \theta||^2$$

$$\leq \mathcal{L}(\theta) + \nabla \mathcal{L}(\theta)^T(\theta^+ - \theta) + \frac{1}{2}L||\theta^+ - \theta||^2$$

Now, we can plug in the gradient projection technique (Equation 15) by letting $\theta^+ = \theta - t \cdot (\mathbf{g} - \frac{\mathbf{g_1} \cdot \mathbf{g_2}}{||\mathbf{g_1}||^2}\mathbf{g_1} - \frac{\mathbf{g_1} \cdot \mathbf{g_2}}{||\mathbf{g_2}||^2}\mathbf{g_2})$. We then get:

$$\mathcal{L}(\theta^+) \leq \mathcal{L}(\theta) + t \cdot \mathbf{g}^T(-\mathbf{g} + \frac{\mathbf{g_1} \cdot \mathbf{g_2}}{||\mathbf{g_1}||^2}\mathbf{g_1} + \frac{\mathbf{g_1} \cdot \mathbf{g_2}}{||\mathbf{g_2}||^2}\mathbf{g_2}) + \frac{1}{2}Lt^2||\mathbf{g} - \frac{\mathbf{g_1} \cdot \mathbf{g_2}}{||\mathbf{g_1}||^2}\mathbf{g_1} - \frac{\mathbf{g_1} \cdot \mathbf{g_2}}{||\mathbf{g_2}||^2}\mathbf{g_2}||^2$$

(Expanding, using the identity $\mathbf{g} = \mathbf{g_1} + \mathbf{g_2}$)

$$= \mathcal{L}(\theta) + t\left(-||\mathbf{g_1}||^2 - ||\mathbf{g_2}||^2 + \frac{(\mathbf{g_1} \cdot \mathbf{g_2})^2}{||\mathbf{g_1}||^2} + \frac{(\mathbf{g_1} \cdot \mathbf{g_2})^2}{||\mathbf{g_2}||^2}\right) + \frac{1}{2}Lt^2||\mathbf{g_1} + \mathbf{g_2}$$
$$- \frac{\mathbf{g_1} \cdot \mathbf{g_2}}{||\mathbf{g_1}||^2}\mathbf{g_1} - \frac{\mathbf{g_1} \cdot \mathbf{g_2}}{||\mathbf{g_2}||^2}\mathbf{g_2}||^2$$

(Expanding further and re-arranging terms)

$$= \mathcal{L}(\theta) - (t - \frac{1}{2}Lt^2)(||\mathbf{g_1}||^2 + ||\mathbf{g_2}||^2 - \frac{(\mathbf{g_1} \cdot \mathbf{g_2})^2}{||\mathbf{g_1}||^2} - \frac{(\mathbf{g_1} \cdot \mathbf{g_2})^2}{||\mathbf{g_2}||^2})$$
$$- Lt^2(\mathbf{g_1} \cdot \mathbf{g_2} - \frac{(\mathbf{g_1} \cdot \mathbf{g_2})^2}{||\mathbf{g_1}||^2||\mathbf{g_2}||^2}\mathbf{g_1} \cdot \mathbf{g_2})$$

(Using the identity $\cos(\phi_{12}) = \frac{\mathbf{g_1} \cdot \mathbf{g_2}}{||\mathbf{g_1}||||\mathbf{g_2}||}$)

$$= \mathcal{L}(\theta) - (t - \frac{1}{2}Lt^2)[(1 - \cos^2(\phi_{12}))||\mathbf{g_1}||^2 + (1 - \cos^2(\phi_{12}))||\mathbf{g_2}||^2]$$
$$- Lt^2(1 - \cos^2(\phi_{12}))||\mathbf{g_1}||||\mathbf{g_2}||\cos(\phi_{12}) \tag{21}$$

(Note that $\cos(\phi_{12}) < 0$ so the final term is non-negative)

Using $t \leq \frac{1}{L}$, we know that $-(1 - \frac{1}{2}Lt) = \frac{1}{2}Lt - 1 \leq \frac{1}{2}L(1/L) - 1 = \frac{-1}{2}$ and $Lt^2 \leq t$.

Plugging this into the last expression above, we can conclude the following:

$$\mathcal{L}(\theta^+) \leq \mathcal{L}(\theta) - \frac{1}{2}t[(1 - \cos^2(\phi_{12}))||\mathbf{g_1}||^2 + (1 - \cos^2(\phi_{12}))||\mathbf{g_2}||^2]$$
$$- t(1 - \cos^2(\phi_{12}))||\mathbf{g_1}||||\mathbf{g_2}||\cos(\phi_{12})$$
$$= \mathcal{L}(\theta) - \frac{1}{2}t(1 - \cos^2(\phi_{12}))[||\mathbf{g_1}||^2 + 2||\mathbf{g_1}||||\mathbf{g_2}||\cos(\phi_{12}) + ||\mathbf{g_2}||^2]$$
$$= \mathcal{L}(\theta) - \frac{1}{2}t(1 - \cos^2(\phi_{12}))[||\mathbf{g_1}||^2 + 2\mathbf{g_1} \cdot \mathbf{g_2} + ||\mathbf{g_2}||^2]$$
$$= \mathcal{L}(\theta) - \frac{1}{2}t(1 - \cos^2(\phi_{12}))||\mathbf{g_1} + \mathbf{g_2}||^2$$
$$= \mathcal{L}(\theta) - \frac{1}{2}t(1 - \cos^2(\phi_{12}))||\mathbf{g}||^2$$

If $\cos(\phi_{12}) > -1$, then $\frac{1}{2}t(1 - \cos^2(\phi_{12}))||\mathbf{g}||^2$ will always be positive unless $\mathbf{g} = 0$. This inequality implies that the objective function value strictly decreases with each iteration where $\cos(\phi_{12}) > -1$.

Hence repeatedly applying gradient projection technique process can either reach the optimal value $\mathcal{L}(\theta) = \mathcal{L}(\theta^*)$ or $\cos(\phi_{12}) = -1$, in which case $\frac{1}{2}t(1 - \cos^2(\phi_{12}))||\mathbf{g}||^2 = 0$. Note that this result only holds when we choose $t$ to be small enough, i.e. $t \leq \frac{1}{L}$.

$\square$

Table 6: The similarity and corresponding performance in the Ant-Dir-OOD environments.

|  | Expert distribution | Prompt-Tuning DT | Prompt Diffuser |
|---|---|---|---|
| Similarity | 1.0 | 0.9514 | 0.9092 |
| Performance | $526.0 \pm 1.5$ | $540 \pm 8.7$ | $546.8 \pm 9.3$ |

## H  PROMPT DIVERSITY ASSESSMENT

We have conducted both quantitative and qualitative analyses to evaluate the diversity of the generated prompts, comparing them with baseline methods to underscore their quality and variation.

**Quantitative Analysis.**  Our quantitative analysis utilizes the Centered Kernel Alignment (CKA) metric (Kornblith et al., 2019), a sophisticated measure widely recognized in machine learning and deep learning research. CKA evaluates the similarity between two sets of features or representations by comparing the alignment of their centered kernel matrices. It produces a single value that quantifies this similarity, with higher values indicating greater resemblance.

To provide a concrete example, we examine the results from the Ant-dir-OOD environment. Here, CKA allows us to quantitatively assess the diversity of prompts generated by different methods. By comparing the CKA scores across various methods, we can determine the extent to which our model generates diverse prompts in comparison to other approaches. As depicted in Table 6, it is important to note that similarity, in isolation, is not a definitive measure of performance. However, the lower similarity score achieved by our Prompt Diffuser, relative to baseline methods, indicates a significant divergence from the established expert distribution. Notably, this divergence does not compromise performance; in fact, our Prompt Diffuser attains the highest performance metrics among the compared methods. This outcome not only demonstrates the effectiveness of our method but also highlights its capability to generate diverse prompts that effectively enhance performance in offline RL scenarios.

**Qualitative Analysis.**  Alongside the quantitative approach, we also conduct a qualitative analysis. This involves visually inspecting and evaluating the prompts generated by our Prompt Diffuser and comparing them with those generated by baseline methods. This qualitative assessment provides a more intuitive understanding of the diversity in the generated prompts.

To visualize the diversity of the prompts, we utilize the t-SNE method for dimensionality reduction and present these prompts in a two-dimensional plane. This visual representation is shown in Figure 5. The t-SNE visualization corroborates our quantitative findings, showing that both the Prompt-Tuning DT and Prompt Diffuser deviate from the original distribution to different extents, leading to performance improvements.

In conclusion, through a comprehensive combination of quantitative (CKA metric) and qualitative analyses, we have rigorously evaluated the diversity of the prompts generated by our model. The results affirm the quality and variation in these prompts, underscoring the effectiveness of our approach in generating diverse and useful prompts for offline RL.

## I  MORE ABLATION STUDY

**Out-of-Distribution Test.**  To assess the OOD generalization, we have conducted tests in the Ant-Dir environment with a specific focus on scenarios that fall outside the distribution of our training set. The training and testing sets for this OOD evaluation are detailed as Table 7.

The results of this OOD evaluation are presented in Table 9. We restrict our comparison to PDT (using expert prompts), Prompt-Tuning DT, and our Prompt Diffuser. The outcomes indicate that the Prompt Diffuser demonstrates notable efficacy even in these OOD tasks. These findings suggest that our Prompt Diffuser not only adapts to and performs well in environments it was trained on but also exhibits commendable generalization to novel, out-of-distribution tasks. This robustness

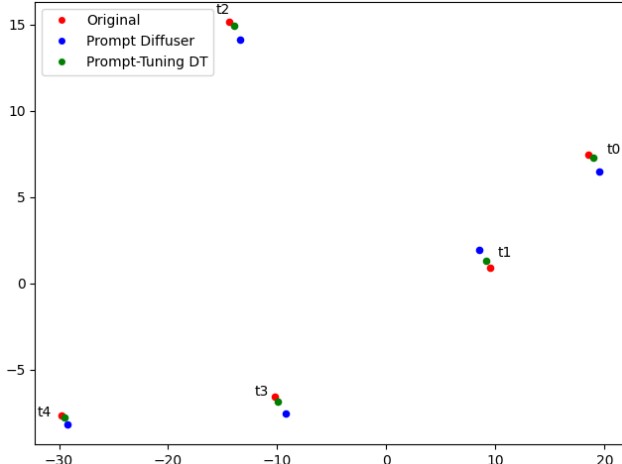

Figure 5: The t-SNE visualization of different prompts in the Ant-Dir-OOD environment.

Table 7: Training and testing task indexes when testing the generalization ability in meta-RL OOD tasks.

| Ant-Dir-OOD | |
| --- | --- |
| Training set of size 8 | [8, 13, 16, 20, 22, 26, 32, 37] |
| Testing set of size 3 | [1, 4, 41] |

in OOD scenarios is a crucial aspect of offline RL and speaks to the effectiveness of our model in diverse and unforeseen environments.

**Prompt Length.** The majority of methods tend to set the prompt length at 5 (Hu et al., 2023b; Xu et al., 2022), encompassing essential information to identify the distribution of the given dataset while avoiding direct imitation. Additionally, opting for shorter prompts is more practical as longer prompts require additional manual effort.

We have also conducted an ablation study focusing on the effects of varying prompt sizes, particularly in the context of the Cheetah-vel environment. This study aims to find an optimal balance between the richness of information provided by longer prompts and the efficiency of the prompt generation process. Our study explores four different prompt length settings, carefully examining their impact on the performance of our Prompt Diffuser model. The central consideration here is the trade-off between longer prompts, which offer more detailed target tasks information, and the increased computational burden they impose on the diffusion loss, particularly due to the need for more denoising iterations to accurately model the complex, extended prompt distributions. The results in Table 8 indicate that while longer prompts initially contribute to improved performance by providing more detailed information, there is a tipping point beyond which the quality of prompt generation begins to decline. This decline is particularly noticeable when the length of the prompt substantially increases, yet the number of denoising iterations remains relatively low, a constraint imposed to maintain processing speed. Interestingly, when the number of denoising iterations (N) is increased for longer prompts (length 40), there is a noticeable improvement in performance. This suggests that the challenge in generating high-quality prompts for longer sequences can be partially mitigated by allowing more iterations in the denoising process.

**Zero-Shot Setting.** In the context of zero-shot settings, where no information about unseen tasks is available, and the model cannot resort to task-specific prompts or undergo further fine-tuning with few-shot prompt datasets, traditional prompt tuning and adaptor methods are not applicable. This scenario presents a unique challenge and holds significant research value.

Table 8: The ablation study on the prompt length in the Ant-Dir-OOD environment.

| Prompt Length | Prompt Diffuser |
|---|---|
| 2 | $-41.2 \pm 11.2$ |
| 5 | $-35.3 \pm 2.4$ |
| 10 | $-40.3 \pm 7.2$ |
| 40 | $-45.3 \pm 4.3$ |
| 40 (with denoising number N increase) | $-43.2 \pm 3.3$ |

Table 9: The ablation study on the zero-shot setting in the Ant-Dir-OOD environment.

| Env Setting | Prompt-DT few-shot | Prompt-Tuning DT few-shot | Prompt Diffuser few-shot | Prompt-DT zero-shot | Prompt Diffuser zero-shot |
|---|---|---|---|---|---|
| Ant-Dir-OOD | $526.0 \pm 1.5$ | $540.8 \pm 8.7$ | $546.8 \pm 9.3$ | $52.7 \pm 1.9$ | $329.2 \pm 21.8$ |

To rigorously assess the zero-shot performance, we conduct an OOD evaluation in the Ant-Dir environment. This environment is specifically chosen due to its diverse range of tasks, providing a robust testbed for our model. The performance in the Ant-Dir-OOD environment is indicative of the model's capability in zero-shot settings. We present a comparative analysis of the performance in Table 9. Notably, in the zero-shot setting, the Prompt Diffuser significantly outperforms the traditional Prompt-DT method. This outcome highlights the efficacy of our model in adapting to new tasks even without prior exposure or task-specific tuning. However, it is important to note that our model still falls considerably behind the performance achieved in the few-shot settings.

