# OpenReview forum: "Prompt Tuning with Diffusion for Few-Shot Pre-trained Policy Generalization"
_ICLR.cc/2024/Conference — Submitted to ICLR 2024_

### Official Review · Reviewer_GaVK · 2023-10-30

**Soundness:** 3 good
**Presentation:** 3 good
**Contribution:** 2 fair
**Rating:** 5
**Confidence:** 4

**Summary:**

This paper proposes a novel approach called Prompt Diffuser to prompt-tuning for few-shot pre-trained policy generalization in meta-RL tasks. The approach eliminates the reliance on initial prompts and instead generates high-quality prompts from random noise using a conditional diffusion model. The authors conduct experiments to demonstrate the effectiveness of this approach and compare it to other prompt-tuning techniques. The results show that the Prompt Diffuser outperforms other methods regarding sample efficiency and generalization performance.

**Strengths:**

> To my knowledge, this paper presents a new approach to overcome the limitations of prompt-tuning for PromptDT.

> Based on the experimental results, it is evident that the Prompt Diffuser outperforms the previous method in terms of performance.

**Weaknesses:**

> While the experimental performance of the algorithm is commendable, this paper primarily represents a combination of existing methods and needs more substantial innovation.

> The motivation of the paper needs some clarification. In the abstract, the authors emphasize the limitations of previous prompt-tuning methods when applied to previously **unseen** tasks, highlighting the need for enhanced generalization and diversity in generative models. There appears to be a discrepancy in this viewpoint, as the subsequent sections of the paper emphasize the **accuracy** of generating prompts within the existing prompt distribution. However, the diversity and the accuracy seem to be a trade-off for the generation.

**Questions:**

> Could the authors clarify whether Prompt Diffuser should output prompts that are more generalized for previously unseen tasks or more precise within the existing prompt distribution?

> Could the authors visualize or show the similarity between the generated prompts from Prompt Diffuser and the ground truth prompts under unseen target tasks?

> Is there any reason for using DM rather than DT for prompt generation? Can the prompt generation be integrated into DT directly?

---

> ### Author Response · Authors · 2023-11-20
>
> ### Q1
> > While the experimental performance of the algorithm is commendable, this paper primarily represents a combination of existing methods and needs more substantial innovation.
>
> Thanks for your comment. We appreciate the opportunity to further elucidate the innovative aspects of our work, which we believe significantly advance the field of prompt-tuning in RL.
>
> + **Significance of Prompt Tuning in RL**: In the current landscape of large-scale models, fine-tuning represents a crucial research paradigm. The adaptability and efficiency of fine-tuning methodologies directly impact the applicability and performance of these expansive models across various domains. While prompt tuning has been previously explored in fields like NLP, its application and phenomena in the RL domain remain largely uncharted territory. Extending prompt tuning to RL not only broadens its scope but also provides an opportunity to explore its efficacy and challenges in a new context. This exploration constitutes a significant contribution to the field, as it opens up a fresh avenue for applying well-established concepts in novel scenarios.
>
> + **Innovative Application of Conditional Generative Modeling**: As demonstrated in Section 3.3, high-quality prompts play a crucial role in enhancing the performance of RL pre-trained models. However, traditional prompt tuning methods often struggle to achieve optimal results and tend to remain in sub-optimal domains due to the nature of RL formulation. Our approach represents a novel adaptation of prompt-tuning, conceptualized as a form of conditional generative modeling. This is a significant shift from the conventional prompt-tuning methods prevalent in natural language processing (NLP). By applying this concept to the RL domain, we introduce a methodology that goes beyond the traditional boundaries of prompt initialization techniques. This adaptation is a noteworthy innovation, as it leverages the generative capabilities of models to produce prompts from a state of random noise, thereby addressing the limitations inherent in traditional methods.
> + **Empirical Validation and Ablation Studies**: The empirical validation of our approach is comprehensive. We have conducted a series of rigorous ablation studies, comparative analyses with baseline methods, and detailed scenario-based evaluations. These efforts collectively demonstrate the practical relevance and applicability of our method across various RL settings. This thorough empirical investigation not only validates the performance of our model but also provides insights into its adaptability and efficacy in diverse scenarios.
> + **Theoretical Contributions**: In addition to empirical validation, our work also makes theoretical contributions to the understanding of prompt-tuning and generative models in the context of RL. These contributions, which we have elaborated on during the rebuttal stage as supplementary material, offer a deeper understanding of the interplay between prompt-tuning, generative modeling, and RL. They provide a foundation for further research and development in this area, highlighting the potential for innovative applications and advancements.
>
> In conclusion, our work represents a substantial leap in the application of prompt-tuning techniques within the RL domain. The integration of conditional generative modeling, along with our comprehensive empirical and theoretical contributions, underscores the innovative nature of our research. We firmly believe that our approach not only extends the current understanding of prompt-tuning in RL but also opens avenues for future exploration and development in this exciting field.

---

> > ### Author Response · Authors · 2023-11-20
> >
> > ### Q2
> > > Could the authors clarify whether Prompt Diffuser should output prompts that are more generalized for previously unseen tasks or more precise within the existing prompt distribution?
> >
> > Sorry for the confusion. These goals are not mutually exclusive but rather complementary in our approach. By incorporating a few-shot dataset of prompts related to unseen tasks, the Prompt Diffuser initially models this given prompt distribution through diffusion loss. This process ensures that the generated prompts align with the existing distribution, which is instrumental in facilitating generalization to unseen tasks.
> >
> > However, as elucidated in Section 4.2 of our paper, relying solely on diffusion loss causes the generated prompt distribution $p$ to converge towards the provided prompt dataset $q$ about unseen tasks. While this convergence is beneficial, it inherently limits the prompts to the performance level of the behavior trajectories in the offline dataset $D$. In scenarios where access to expert prompt distribution is available (as seen in Prompt-DT-Expert results), this method can yield satisfactory performance. However, challenges arise when the provided distribution significantly deviates from expert quality (e.g., Prompt-DT-Random) or when the expert distribution itself is suboptimal (as observed in the MW reach-v2 environment).
> >
> > To address these challenges, additional downstream task guidance is integrated. This guidance serves to enhance the quality of the prompts, especially when the provided dataset is less than ideal. The incorporation of downstream task guidance not only maintains proximity to the existing prompt distribution but also elevates the quality of the generated prompts by the downstream task loss, thereby boosting their effectiveness in unseen tasks.
> >
> > In summary, the primary objective of the strategies implemented is to enhance the model's generalizability across novel tasks.  The $L_{DM}$ loss plays a crucial role in approximating the distribution of the existing dataset, setting a foundational benchmark for the quality of the generated prompts. Simultaneously, the introduction of downstream task guidance acts as a lever to elevate the potential quality of these prompts. This dual-pronged approach ensures that our methodology maintains consistent and robust performance, even when faced with variations in the quality of prompt datasets. (Section 5.4)

---

> ### Author Response · Authors · 2023-11-20
>
> ### Q3
> > Could the authors visualize or show the similarity between the generated prompts from Prompt Diffuser and the ground truth prompts under unseen target tasks?
>
> Thanks for this suggestion. We have conducted both quantitative and qualitative analyses to evaluate the diversity of the generated prompts, comparing them with baseline methods to underscore their quality and variation.
>
> Quantitative Analysis Using CKA Metric:
>
> + Our quantitative analysis utilizes the Centered Kernel Alignment (CKA) metric [1], a sophisticated measure widely recognized in machine learning and deep learning research. CKA evaluates the similarity between two sets of features or representations by comparing the alignment of their centered kernel matrices. It produces a single value that quantifies this similarity, with higher values indicating greater resemblance.
> + To provide a concrete example, we examine the results from the Ant-dir-OOD environment. Here, CKA allows us to quantitatively assess the diversity of prompts generated by different methods. By comparing the CKA scores across various methods, we can determine the extent to which our model generates diverse prompts in comparison to other approaches.
> + As depicted in the following table, it is important to note that similarity, in isolation, is not a definitive measure of performance. However, the lower similarity score achieved by our Prompt Diffuser, relative to baseline methods, indicates a significant divergence from the established expert distribution. Notably, this divergence does not compromise performance; in fact, our Prompt Diffuser attains the highest performance metrics among the compared methods. This outcome not only demonstrates the effectiveness of our method but also highlights its capability to generate diverse prompts that effectively enhance performance in offline RL scenarios.
>
>
>
> |             | Expert distribution | Prompt-Tuning DT[2] | Prompt Diffuser |
> | ----------- | ------------------- | ---------------- | --------------- |
> | Similarity  | 1.0                 | 0.9514           | 0.9092          |
> | Performance | 526.0 $\pm$ 1.5     | 540 $\pm$ 8.7    | 546.8 $\pm$ 9.3 |
>
>
> Qualitative Analysis:
> + Alongside the quantitative approach, we also conduct a qualitative analysis. This involves visually inspecting and evaluating the prompts generated by our Prompt Diffuser and comparing them with those generated by baseline methods. This qualitative assessment provides a more intuitive understanding of the diversity in the generated prompts.
> + To visualize the diversity of the prompts, we utilize the t-SNE method for dimensionality reduction and present these prompts in a two-dimensional plane. This visual representation is detailed in the Appendix H of our updated draft. The t-SNE visualization corroborates our quantitative findings, showing that both the Prompt-Tuning DT and Prompt Diffuser deviate from the original distribution to different extents, leading to performance improvements.
>
> In conclusion, through a comprehensive combination of quantitative (CKA metric) and qualitative analyses, we have rigorously evaluated the diversity of the prompts generated by our model. The results affirm the quality and variation in these prompts, underscoring the effectiveness of our approach in generating diverse and useful prompts for offline RL.

---

> ### Author Response · Authors · 2023-11-20
>
> ### Q4
> > Is there any reason for using DM rather than DT for prompt generation? Can the prompt generation be integrated into DT directly?
>
> Thanks for the comment. The rationale for this choice is grounded in the distinct capabilities of these models, which we elaborate upon as follows:
> + DMs are renowned for their proficiency in generating high-quality data samples, an essential trait for effective prompt generation in reinforcement learning (RL). In our research, we explored other generative models, such as Variational Autoencoders (VAE), alongside DMs. The comparative results, presented in Appendix Table 5, suggest that the inherent attributes of the generative model significantly influence performance. Specifically, the capacity of DMs to produce in-distribution outcomes and cater to personalized conditions is crucial for our method. This capability has enabled us to reconceptualize prompt tuning within the framework of generative modeling.
> + DT, fundamentally a discriminative model, is optimized for decision-making based on pre-existing prompts. It relies on these prompts to extract information pertinent to unseen tasks. We experimented with treating the prompt trajectory as part of the input, training it with $L_{DT}$ loss, and then using DT to generate prompts at the outset of the inference stage. As depicted in the table below, the suboptimal performance in this setup can be attributed to the following factors:
>     + Limited Generative Capability: DT’s core strength lies in decision-making informed by given prompts. However, its capacity to generate new, high-quality prompts is inherently limited. This shortfall can result in less effective prompts that may not sufficiently guide the model in unfamiliar scenarios.
>     + Complexity in Training Dynamics: Integrating prompt generation with decision-making in DT adds complexity to the training dynamics. Balancing these two aspects effectively can be challenging, potentially leading to compromises in the performance of either or both tasks.
>
>
> | Env | Prompt Diffuser | All-in-DT |
> | -------- | -------- | -------- |
> | Ant-Dir-OOD     | 546.8 $\pm$ 9.3     | 479.9 $\pm$ 5.8     |
>
>
> In conclusion, while theoretically feasible, the integration of prompt generation into DT presents several practical challenges. These include the limitations of DT in generative tasks, its reliance on the quality of initial prompts, and the complexities introduced in training dynamics. Consequently, our decision to utilize DMs for prompt generation is informed by their superior generative capabilities and adaptability to diverse scenarios, which are pivotal for effective and robust performance in RL settings.
>
> ### Reference
>
> [1] Simon, Kornblith, et al. "Similarity of neural network representations revisited." ICML 2019.
>
> [2] Hu, Shengchao, et al. "Prompt-tuning decision transformer with preference ranking." arXiv preprint arXiv:2305.09648, 2023.

---

> > ### Author Response · Authors · 2023-11-23
> >
> > ## We anticipate your feedback!
> >
> > Dear Reviewer GaVK,
> >
> > The authors greatly appreciate your time and effort in reviewing this submission, and eagerly await your response. We understand you might be quite busy. However, the discussion deadline is approaching, and we have only a few hours left.
> >
> > We have provided detailed responses to every one of your concerns/questions. Please help us to review our responses once again and kindly let us know whether they fully or partially address your concerns and if our explanations are in the right direction.
> >
> > Best Regards,
> >
> > The authors of Submission 3141

---

### Official Review · Reviewer_zrXj · 2023-11-01

**Soundness:** 3 good
**Presentation:** 3 good
**Contribution:** 3 good
**Rating:** 8
**Confidence:** 3

**Summary:**

the author propose Prompt Diffuser (PD), a generative module to augment the prompt tuning decision transformer for reinforcement learning. The authors train a reward-to-go conditioned generative model in order to obtain a better initialization of the prompt. The author provide empirical evidence showing its effectiveness on multiple benchmarks and perform proper ablations.

**Strengths:**

This is a good paper because it reveals the important insight about prompt turning in RL, which is the importance of visualization. The movitating example in Fig2 is very helpful to establish the hypothesis. The choice of using a conditional diffusion model is well motivated. The empircal results are solid, and the ablation in Table 2 show the effect of the better initialization from the diffusion model.

**Weaknesses:**

I might miss this, but I would encourage the author to also show the OOD generalization to novel environments as is done in the PDT paper section 6.4 to make the results stronger?

More results on how the diffusion model generalize to unseen reward-to-go?

Any visualization on the diversity of generated prompts/trajectories?

**Questions:**

See above

---

> ### Author Response · Authors · 2023-11-20
>
> ### Q1
> > I would encourage the author to also show the OOD generalization to novel environments as is done in the PDT paper section 6.4 to make the results stronger?
>
> Thanks for your valuable suggestion. We acknowledge the importance of such an evaluation to strengthen the results and validate the robustness of our model in novel environments.
>
> To assess the OOD generalization, we have conducted tests in the Ant-Dir environment with a specific focus on scenarios that fall outside the distribution of our training set. The training and testing sets for this OOD evaluation are detailed as follows:
>
> | Environment | Set Description        | Task IDs                        |
> | ------- | ---------------------- | ------------------------------- |
> | Ant-Dir | Training set of size 8 | [8, 13, 16, 20, 22, 26, 32, 37] |
> | Ant-Dir | Testing set of size 3  | [1, 4, 41]                      |
>
> The results of this OOD evaluation are presented below. Due to time constraints, we restrict our comparison to PDT (using expert prompts)[2], Prompt-Tuning DT[3], and our Prompt Diffuser. The outcomes indicate that the Prompt Diffuser demonstrates notable efficacy even in these OOD tasks:
>
> | Env         | Prompt-DT-Expert | Prompt-Tuning DT | Prompt Diffuser |
> | ----------- | ---------------- | ---------------- | --------------- |
> | Ant-Dir-OOD | 526.0 $\pm$ 1.5  | 540.8 $\pm$ 8.7  | 546.8 $\pm$ 9.3 |
>
> These findings suggest that our Prompt Diffuser not only adapts to and performs well in environments it was trained on but also exhibits commendable generalization to novel, out-of-distribution tasks. This robustness in OOD scenarios is a crucial aspect of offline RL and speaks to the effectiveness of our model in diverse and unforeseen environments.
>
> ### Q2
> > More results on how the diffusion model generalize to unseen reward-to-go?
>
> Thanks for the comment. To address your concern, we have conducted analyses under two distinct scenarios:
>
> **Scenario 1: Inference with Altered Return-to-Go Tokens**:
>
> In this setup, while the Prompt Diffuser is fine-tuned using few-shot prompt datasets, during the inference stage, we modify the return-to-go tokens. Instead of using the original rtg from the given datasets, we add Gaussian noise to these tokens, thereby presenting unseen return-to-go values to the model. The performance under this scenario is outlined below:
>
> | Env         | Prompt-DT-Expert | Prompt Diffuser | Prompt Diffuser with unseen rtg |
> | ----------- | ---------------- | --------------- | ------------------------------- |
> | Ant-Dir-OOD | 526.0 $\pm$ 1.5  | 546.8 $\pm$ 9.3 |       538 $\pm$ 8.6       |
>
> Despite the Prompt Diffuser not having prior exposure to these altered return-to-go (rtg) values, it can still generate effective prompts that guide the model to perform competently in the Ant-Dir-OOD environment. This outcome underscores the model's resilience and its ability to adapt to variations in task-specific parameters, even in the absence of direct prior exposure to such modified conditions. This ability may be attributed to the robustness of generative modeling and  the diverse set of scenarios and rtg values in the training datasets.
>
> **Scenario 2: Zero-Shot Setting with Unseen Reward-to-Go and Tasks:**
>
> In a more stringent setting, we evaluate the Prompt Diffuser under zero-shot conditions, where it lacks access to target prompt datasets for reference. This scenario necessitates handling not only unseen reward-to-go but also previously unseen tasks. The results of this rigorous test are as follows:
>
> | Env         | Prompt-DT-zero-shot | Prompt Diffuser zero-shot |
> | ----------- | ------------------- | ------------------------- |
> | Ant-Dir-OOD | 52.7 $\pm$ 1.9      | 329.2 $\pm$ 21.8          |
>
> Notably, in the zero-shot setting, the Prompt Diffuser significantly outperforms traditional Prompt-DT, underscoring its ability to adapt to new tasks and rtg values without prior task-specific tuning or exposure. However, it is crucial to acknowledge that the performance in zero-shot settings still trails behind what is achieved in few-shot scenarios.
>
> These findings demonstrate the versatility and adaptability of our Prompt Diffuser in handling unseen rtg values and tasks. The model's performance in both altered rtg and zero-shot settings highlights its robust generalization capabilities, an essential feature for RL models operating in dynamic and unpredictable environments.

---

> ### Author Response · Authors · 2023-11-20
>
> ### Q3
> > Any visualization on the diversity of generated prompts/trajectories?
>
> Thanks for this suggestion. We have conducted both quantitative and qualitative analyses to evaluate the diversity of the generated prompts, comparing them with baseline methods to underscore their quality and variation.
>
> Quantitative Analysis Using CKA Metric:
>
> + Our quantitative analysis utilizes the Centered Kernel Alignment (CKA) metric [1], a sophisticated measure widely recognized in machine learning and deep learning research. CKA evaluates the similarity between two sets of features or representations by comparing the alignment of their centered kernel matrices. It produces a single value that quantifies this similarity, with higher values indicating greater resemblance.
> + To provide a concrete example, we examine the results from the Ant-dir-OOD environment. Here, CKA allows us to quantitatively assess the diversity of prompts generated by different methods. By comparing the CKA scores across various methods, we can determine the extent to which our model generates diverse prompts in comparison to other approaches.
> + As depicted in the following table, it is important to note that similarity, in isolation, is not a definitive measure of performance. However, the lower similarity score achieved by our Prompt Diffuser, relative to baseline methods, indicates a significant divergence from the established expert distribution. Notably, this divergence does not compromise performance; in fact, our Prompt Diffuser attains the highest performance metrics among the compared methods. This outcome not only demonstrates the effectiveness of our method but also highlights its capability to generate diverse prompts that effectively enhance performance in offline RL scenarios.
>
>
>
> |             | Expert distribution | Prompt-Tuning DT | Prompt Diffuser |
> | ----------- | ------------------- | ---------------- | --------------- |
> | Similarity  | 1.0                 | 0.9514           | 0.9092          |
> | Performance | 526.0 $\pm$ 1.5     | 540 $\pm$ 8.7    | 546.8 $\pm$ 9.3 |
>
>
> Qualitative Analysis:
> + Alongside the quantitative approach, we also conduct a qualitative analysis. This involves visually inspecting and evaluating the prompts generated by our Prompt Diffuser and comparing them with those generated by baseline methods. This qualitative assessment provides a more intuitive understanding of the diversity in the generated prompts.
> + To visualize the diversity of the prompts, we utilize the t-SNE method for dimensionality reduction and present these prompts in a two-dimensional plane. This visual representation is detailed in the Appendix H of our updated draft. The t-SNE visualization corroborates our quantitative findings, showing that both the Prompt-Tuning DT and Prompt Diffuser deviate from the original distribution to different extents, leading to performance improvements.
>
> In conclusion, through a comprehensive combination of quantitative (CKA metric) and qualitative analyses, we have rigorously evaluated the diversity of the prompts generated by our model. The results affirm the quality and variation in these prompts, underscoring the effectiveness of our approach in generating diverse and useful prompts for offline RL.
>
> ### Reference
>
> [1] Simon, Kornblith, et al. "Similarity of neural network representations revisited." ICML 2019.
>
> [2] Xu, Mengdi, et al. "Prompting decision transformer for few-shot policy generalization." ICML 2022.
>
> [3] Hu, Shengchao, et al. "Prompt-tuning decision transformer with preference ranking." arXiv preprint arXiv:2305.09648, 2023.

---

### Official Review · Reviewer_4ueS · 2023-11-02

**Soundness:** 2 fair
**Presentation:** 3 good
**Contribution:** 2 fair
**Rating:** 5
**Confidence:** 3

**Summary:**

The work introduces using generative modeling using a diffusion process to do prompt tuning where the prompts are generated using random noise. The method uses signals from downstream tasks to generate better prompts, thus gaining advantage over methods which do not use this form of signal propagation.

**Strengths:**

1. The method provides an interesting way to use diffusion methods for few-shot policy learning.
2. By using downstream task information and gradient guidance, the task can achieve good performance as compared to baseline methods.
3. The work is an interesting direction for prompt-tuning methods.

**Weaknesses:**

1. The work doesn't discuss the diversity of the prompts generated. An analysis, quantitative or qualitative, can help understand if there is enough variation in the generated prompts and their comparison with baseline methods.
2. The length of the prompts is considered small. An ablation over the effect of prompt sizes can be important.
3. The effect of downstream tasks seems to be an important factor in contributing towards the success of the method. An ablation on the effect of this vs the rest of the method will be useful - one example is - if the downstream task is restricted in some manner.
4. Although the method considers few-shot settings, an additional examination of the zero-shot setting can be useful to evaluate the transfer capabilities and analyze how tried the generated prompts are to the downstream method.

**Questions:**

1. How is the quality of the prompts generated? Is there enough diversity?
2. Have different prompt lengths been tried and what is their impact?
3. What is the impact of quality and size of downstream tasks, and zero-shot performance?

---

> ### Author Response · Authors · 2023-11-20
>
> ### Q1
> > The work doesn't discuss the diversity of the prompts generated. An analysis, quantitative or qualitative, can help understand if there is enough variation in the generated prompts and their comparison with baseline methods. How is the quality of the prompts generated? Is there enough diversity?
>
> Thanks for this suggestion. We have conducted both quantitative and qualitative analyses to evaluate the diversity of the generated prompts, comparing them with baseline methods to underscore their quality and variation.
>
> Quantitative Analysis Using CKA Metric:
>
> + Our quantitative analysis utilizes the Centered Kernel Alignment (CKA) metric [1], a sophisticated measure widely recognized in machine learning and deep learning research. CKA evaluates the similarity between two sets of features or representations by comparing the alignment of their centered kernel matrices. It produces a single value that quantifies this similarity, with higher values indicating greater resemblance.
> + To provide a concrete example, we examine the results from the Ant-dir-OOD environment. Here, CKA allows us to quantitatively assess the diversity of prompts generated by different methods. By comparing the CKA scores across various methods, we can determine the extent to which our model generates diverse prompts in comparison to other approaches.
> + As depicted in the following table, it is important to note that similarity, in isolation, is not a definitive measure of performance. However, the lower similarity score achieved by our Prompt Diffuser, relative to baseline methods, indicates a significant divergence from the established expert distribution. Notably, this divergence does not compromise performance; in fact, our Prompt Diffuser attains the highest performance metrics among the compared methods. This outcome not only demonstrates the effectiveness of our method but also highlights its capability to generate diverse prompts that effectively enhance performance in offline RL scenarios.
>
>
>
> |             | Expert distribution | Prompt-Tuning DT[3] | Prompt Diffuser |
> | ----------- | ------------------- | ---------------- | --------------- |
> | Similarity  | 1.0                 | 0.9514           | 0.9092          |
> | Performance | 526.0 $\pm$ 1.5     | 540 $\pm$ 8.7    | 546.8 $\pm$ 9.3 |
>
>
> Qualitative Analysis:
> + Alongside the quantitative approach, we also conduct a qualitative analysis. This involves visually inspecting and evaluating the prompts generated by our Prompt Diffuser and comparing them with those generated by baseline methods. This qualitative assessment provides a more intuitive understanding of the diversity in the generated prompts.
> + To visualize the diversity of the prompts, we utilize the t-SNE method for dimensionality reduction and present these prompts in a two-dimensional plane. This visual representation is detailed in the Appendix H of our updated draft. The t-SNE visualization corroborates our quantitative findings, showing that both the Prompt-Tuning DT and Prompt Diffuser deviate from the original distribution to different extents, leading to performance improvements.
>
> In conclusion, through a comprehensive combination of quantitative (CKA metric) and qualitative analyses, we have rigorously evaluated the diversity of the prompts generated by our model. The results affirm the quality and variation in these prompts, underscoring the effectiveness of our approach in generating diverse and useful prompts for offline RL.
>
> ### Q2
> > The length of the prompts is considered small. An ablation over the effect of prompt sizes can be important. Have different prompt lengths been tried and what is their impact?
>
> Thanks for the comment. The majority of methods tend to set the prompt length at 5 [2,3], encompassing essential information to identify the distribution of the given dataset while avoiding direct imitation. Additionally, opting for shorter prompts is more practical as longer prompts require additional manual effort.
>
> We have also conducted an ablation study focusing on the effects of varying prompt sizes, particularly in the context of the Cheetah-vel environment. This study aims to find an optimal balance between the richness of information provided by longer prompts and the efficiency of the prompt generation process.
>
> Our study explores four different prompt length settings, carefully examining their impact on the performance of our Prompt Diffuser model. The central consideration here is the trade-off between longer prompts, which offer more detailed target tasks information, and the increased computational burden they impose on the diffusion loss, particularly due to the need for more denoising iterations to accurately model the complex, extended prompt distributions.
>
> | Prompt Length| Prompt Diffuser  |
> | -| -  |
> | 2   | -40.2 $\pm$ 11.2 |
> | 5 | -35.3 $\pm$  2.4 |
> | 10 | -38.3 $\pm$ 7.2  |
> | 40  | -45.3 $\pm$ 4.3  |
> | 40 (with denoising number N increase) | -42.2  $\pm$ 3.3 |

---

> > ### Author Response · Authors · 2023-11-20
> >
> > (Continuing Q2)
> >
> > The results indicate that while longer prompts initially contribute to improved performance by providing more detailed information, there is a tipping point beyond which the quality of prompt generation begins to decline. This decline is particularly noticeable when the length of the prompt substantially increases, yet the number of denoising iterations remains relatively low, a constraint imposed to maintain processing speed. Interestingly, when the number of denoising iterations (N) is increased for longer prompts (length 40), there is a noticeable improvement in performance. This suggests that the challenge in generating high-quality prompts for longer sequences can be partially mitigated by allowing more iterations in the denoising process.
> >
> > In conclusion, this ablation study underscores the complexity and importance of choosing an appropriate prompt length in the context of our Prompt Diffuser model. It highlights the need to balance the informational richness of longer prompts with the computational demands they entail, especially considering the performance implications in various scenarios.
> >
> > ### Q3
> > > The effect of downstream tasks seems to be an important factor in contributing towards the success of the method. An ablation on the effect of this vs the rest of the method will be useful - one example is - if the downstream task is restricted in some manner.
> >
> > Thanks for the comment. We have conducted an extensive ablation study on the effect of Diffusion Guidance, which examines the interplay between diffusion loss and downstream task guidance in our model's formulation. This study includes four distinct formulations:
> > + Equation 18: This formulation exclusively employs diffusion loss to imitate the prompt datasets, effectively removing any downstream task guidance. It serves as a baseline to understand the role of diffusion loss in isolation.
> > + Equation 19: Here, we focus solely on the downstream task guidance, omitting the diffusion loss that typically imitates the prompt datasets. This approach allows us to assess the impact of downstream guidance independently.
> > + Equation 20: Represents a simple linear combination of the diffusion loss and downstream guidance. This formulation tests the efficacy of a straightforward integration of both losses.
> > + Equation 15: Involves using our gradient projection technique to integrate both losses. This sophisticated approach aims to balance the contributions of diffusion loss and downstream task guidance optimally.
> >
> > As illustrated in Figure 3, the effectiveness of downstream task guidance varies depending on the environment:
> > + In environments like Cheetah-vel and MW reach-v2, where downstream loss positively influences performance (as shown in Equation 19), a linear combination of losses (Equation 20) does not consistently outperform the individual losses. Notably, in the Cheetah-vel environment, Equation 20 leads to suboptimal performance. However, our gradient projection technique (Equation 15) effectively alleviates this issue, consistently enhancing performance.
> > + Conversely, in environments like Ant-dir, where the inclusion of downstream loss appears to diminish performance, our gradient projection method maintains effectiveness with relying solely on diffusion loss without performance degradation.
> >
> > These findings suggest that the incorporation of downstream task loss is not universally beneficial and requires careful consideration and design. The gradient projection technique, in particular, stands out for its ability to judiciously balance these losses, thereby enhancing the overall effectiveness of the Prompt Diffuser method in a variety of task environments.

---

> > > ### Author Response · Authors · 2023-11-20
> > >
> > > ### Q4
> > > > What is the impact of quality and size of downstream tasks, and zero-shot performance?
> > >
> > > Thanks for the comment. Our findings, as outlined in following table, demonstrate the robustness of the Prompt Diffuser to variations in the quality and size of training data for downstream tasks. This robustness stems from our unique approach in which the prompt-tuning process is conceptualized as a form of conditional generative modeling, utilizing downstream task loss as an additional guidance mechanism.
> > >
> > > When the quality of downstream task datasets is suboptimal, potentially leading to negative performance impacts, our gradient projection technique plays a crucial role. It ensures that performance is maintained, relying predominantly on diffusion loss to prevent degradation due to inferior data quality. The size of downstream task datasets similarly affects the performance of our model. The primary influence of dataset size and quality lies in determining the effectiveness of the $L_{DT}$ loss: higher quality or larger datasets typically enhance better $L_{DT}$, while in scenarios where $L_{DT}$ is of lower quality, our gradient projection technique effectively shields the model from adverse effects, which maintains effectiveness with relying solely on diffusion loss without performance degradation.
> > >
> > > | Quality | Size | Performance |
> > > | ------- | ---- | ----------- |
> > > | Expert  | 256  | -32.6 $\pm$ 1.3        |
> > > | Medium  | 256  | -34.2 $\pm$ 2.8            |
> > > | Random  | 256  | -33.5 $\pm$ 2.0            |
> > > | Mix     | 32   | -36.5 $\pm$  2.4           |
> > > | Mix     | 64   |  -35.6 $\pm$ 2.1           |
> > > | Mix     | 128  |   -34.2 $\pm$ 2.3          |
> > > | Mix     | 256  |  -35.3 $\pm$ 2.4           |
> > >
> > > In the context of zero-shot settings, where no information about unseen tasks is available, and the model cannot resort to task-specific prompts or undergo further fine-tuning with few-shot prompt datasets, traditional prompt tuning and adaptor methods are not applicable. This scenario presents a unique challenge and holds significant research value.
> > >
> > > To rigorously assess the zero-shot performance, we conduct an out-of-distribution (OOD) evaluation in the Ant-Dir environment. This environment is specifically chosen due to its diverse range of tasks, providing a robust testbed for our model. The training and testing sets for this OOD evaluation are as follows:
> > >
> > > | Environment | Set Description        | Task IDs                        |
> > > | ----------- | ---------------------- | ------------------------------- |
> > > | Ant-Dir     | Training set of size 8 | [8, 13, 16, 20, 22, 26, 32, 37] |
> > > | Ant-Dir     | Testing set of size 3  | [1, 4, 41]                      |
> > >
> > > The performance in the Ant-Dir-OOD environment is indicative of the model's capability in zero-shot settings. We present a comparative analysis of the performance across different setups:
> > >
> > >
> > > | Env         | Prompt-DT-few-shot |  Prompt Diffuser few-shot | Prompt-DT-zero-shot | Prompt Diffuser zero-shot |
> > > | ----------- |  ---------------- | ------------------------ | ------------------- | ------------------------- |
> > > | Ant-Dir-OOD | 526.0 $\pm$ 1.5    |  546.8 $\pm$ 9.3          | 52.7 $\pm$ 1.9      | 329.2 $\pm$ 21.8          |
> > >
> > > Notably, in the zero-shot setting, the Prompt Diffuser significantly outperforms the traditional Prompt-DT method. This outcome highlights the efficacy of our model in adapting to new tasks even without prior exposure or task-specific tuning. However, it is important to note that our model still falls considerably behind the performance achieved in the few-shot settings.
> > >
> > > ### Reference
> > > [1] Simon, Kornblith, et al. "Similarity of neural network representations revisited." ICML 2019.
> > >
> > > [2] Xu, Mengdi, et al. "Prompting decision transformer for few-shot policy generalization." ICML 2022.
> > >
> > > [3] Hu, Shengchao, et al. "Prompt-tuning decision transformer with preference ranking." arXiv preprint arXiv:2305.09648, 2023.

---

> ### Author Response · Authors · 2023-11-23
>
> ## We anticipate your feedback!
>
> Dear Reviewer 4ueS,
>
> Thanks very much for your time and valuable comments.
>
> We understand you might be quite busy. However, the discussion deadline is approaching, and we have only a few hours left.
>
> Would you mind checking our response and confirming whether you have any further questions?
>
> Thanks for your attention.
>
> Best regards,
>
> The authors of submission 3141.

---

### Official Review · Reviewer_V1Dp · 2023-11-09

**Soundness:** 3 good
**Presentation:** 2 fair
**Contribution:** 3 good
**Rating:** 6
**Confidence:** 4

**Summary:**

This paper identifies the generalization challenges in offline reinforcement learning due to the lack of data quantity and quality. The author(s) build a connection with pre-trained large-scale models that share a similar difficulty. Based on that, a new prompt-tuning method using generative modeling is proposed to improve the performance of offline RL.

**Strengths:**

The idea to identify similar difficulties with PLM and align with state-of-the-art technical is innovative.

The paper provides a detailed review and a smooth transition from the existing work of both PLMs and offline RL to the proposed work.

The methodology demonstration is clear and easy to follow.

**Weaknesses:**

The presentation of figures can be improved, e.g., the text in Figure 2 is too small compared to the bar size, so as in Figure 3.

The proposed method needs theoretical support other than empirical analysis to avoid being a mix of heuristic designs.

For experiments, the testing scenarios are limited to answering the three important questions at the beginning of Sec. 5. For example, the results of the ablation study in Figure 3 don’t have a common conclusion and need more analysis. Without comprehensive experiments and insights, It’s hard to generalize the approach to other problems.

**Questions:**

How big is the difference between seen and unseen environments and tasks in the testing cases? As this part of the motivation, the results may be sensitive to the scenario changes. Have you quantified the difference that makes the prompt-tuning useful for offline RL?

The reviewer doubts the effectiveness of using guidance from downstream tasks. Interestingly, Figure 3, which does not have the label for performance metric, doesn’t present the same conclusion for different tasks. Guidance loss may not be useful sometimes. Can you explain why?

When considering the two losses, the gradient projection seems beneficial, but is there a comparison to validate the superiority over a linear combination?

---

> ### Author Response · Authors · 2023-11-20
>
> ### Q1
> > The proposed method needs theoretical support other than empirical analysis to avoid being a mix of heuristic designs.
>
> Thanks for this suggestion! From the perspective of optimization, motivated by [3], we give the following theoretical support for our gradient projection technique for the final performance.
>
> *Assume $L_{DM}$ and $L_{DT}$ are convex and differentiable. Suppose the gradient of $L = L_{DM} + L_{DT}$ is $L$-Lipschitz with $L > 0$. Then, the gradient projection technique with step size $t \leq \frac{1}{L}$ will converge to either (1) a location in the optimization landscape where $\cos(\phi) = -1$ or (2) the optimal value $L(\\theta^\*)$.*
>
> The convergence to a point where $\cos(\phi) = -1$ indicates a scenario where the gradients of $L_{DM}$ and $L_{DT}$ are in opposite directions. This situation typically arises when there is a conflict between the objectives of the two loss functions. In practical implementation, our priority is to ensure that the DM loss converges normally, even if it means sacrificing the convergence of the DT loss to some extent. This is because the $L_{DM}$ loss plays a crucial role in approximating the distribution of the existing dataset, which sets a foundational benchmark for the quality of the generated prompts.
>
> On the other hand, convergence to $L(\theta^*)$ signifies that the optimization process successfully finds the optimal point that minimizes the combined loss. This outcome is the ideal scenario, demonstrating the effectiveness of our gradient projection technique in jointly optimizing both loss functions.
>
> This theoretical framework addresses a critical challenge in combining multiple loss functions, particularly when these functions have potentially conflicting gradients. Our approach ensures stable and effective optimization, even in complex scenarios where balancing multiple objectives is necessary.
>
> For ease of understanding,  we present the detailed proof process below (simplified, see the updated draft for the full version):
>
> We will use the shorthand $|| \cdot ||$ to denote the $L_2$-norm and $\nabla L = \nabla_\theta L$, where $\theta$ is the parameter vector.  let $g_1 = \nabla L_{DM}$, $g_2 = \nabla L_{DT}$, $g = \nabla L = g_1 + g_2$, and $\phi_{12}$ be the angle between $g_1$ and $g_2$.
>
> Our assumption that $\nabla L$ is Lipschitz continuous with constant $L$ implies that $\nabla^2 L(\theta) - LI$ is a negative semi-definite matrix. Using this fact, we can perform a quadratic expansion of $L$ around $L(\theta)$ and obtain the following inequality:
> \begin{align*}
> L(\theta^+) &\leq L(\theta) + \nabla L(\theta)^T (\theta^+ - \theta) + \frac{1}{2} \nabla^2 L(\theta) ||\theta^+ - \theta||^2
> \leq L(\theta) + \nabla L(\theta)^T (\theta^+ - \theta) + \frac{1}{2} L ||\theta^+ - \theta||^2
> \end{align*}
>
> Now, we can plug in the gradient projection technique by letting $\theta^+ = \theta - t\cdot(g - \frac{g_1 \cdot g_2}{||g_1||^2}g_1 - \frac{g_1 \cdot g_2}{||g_2||^2}g_2)$. We then get:
>
> \begin{align*}
> L(\theta^+) &\leq
>      L(\theta) - (t  - \frac{1}{2}Lt^2) [(1 - \cos^2(\phi_{12})) ||g_1||^2 + (1 - \cos^2(\phi_{12})) ||g_2||^2 ]
>     - Lt^2 (1 - \cos^2(\phi_{12})) ||g_1|| ||g_2|| \cos(\phi_{12}) \\
> \end{align*}
>
> Using $t \leq \frac{1}{L}$, we know that $-(1 - \frac{1}{2} Lt) = \frac{1}{2}Lt - 1 \leq \frac{1}{2}L(1/L) - 1 = \frac{-1}{2}$ and $Lt^2 \leq t$.
>
> Plugging this into the expression above and $\cos({\phi_{12}}) < 0$, we can conclude the following:
> \begin{align*}
>     L(\theta^+) &\leq
>      L(\theta) - \frac{1}{2}t (1 - \cos^2(\phi_{12})) \|g\|^2
> \end{align*}
>
> If $\cos(\phi_{12}) > -1$, then $\frac{1}{2}t (1 - \cos^2(\phi_{12})) \|g\|^2$ will always be positive unless $g = 0$. This inequality implies that the objective function
> value strictly decreases with each iteration where $\cos(\phi_{12}) > -1$.
>
> Hence repeatedly applying gradient projection technique process can either reach the optimal value $L(\theta) = L(\theta^*)$ or $\cos(\phi_{12}) = -1$, in which case $\frac{1}{2}t (1 - \cos^2(\phi_{12})) \|g\|^2 = 0$. Note that this result only holds when we choose $t$ to be small enough, i.e. $t \leq \frac{1}{L}$.

---

> > ### Author Response · Authors · 2023-11-20
> >
> > ### Q2
> > > For experiments, the testing scenarios are limited to answering the three important questions at the beginning of Sec. 5. For example, the results of the ablation study in Figure 3 don’t have a common conclusion and need more analysis. Without comprehensive experiments and insights, It’s hard to generalize the approach to other problems.
> >
> > Thanks for the valuable suggestion. In Section 5 of our manuscript, we delve into three critical questions, below we will provide a more comprehensive analysis that will be integrated into the revised draft:
> >
> > (1) **Enhanced Model Generalization through Improved Prompt Generation**: The core of our study is reflected in the main results, where we demonstrate the ability of our model to generate superior prompts, leading to enhanced performance relative to other parameter-efficient tuning techniques.
> >
> > (2) **Robust to the Prompt Dataset Quality**: Addressed in the Ablation on Prompt Initialization, we investigate the influence of prompt dataset quality. Given that our model's exposure to unseen tasks is primarily through few-shot target trajectories, understanding the necessity of high-quality datasets is vital. High-quality datasets often require extensive labor and are time-consuming to compile. However, as shown in Table 2, our Prompt Diffuser can produce high-quality prompts even when fine-tuned with relatively poor random prompt datasets. This finding is significant, indicating our model's robustness and efficiency in diverse scenarios.
> >
> > (3) **Balancing Diffusion Guidance with DDPM Updates**: Our Ablation on Diffusion Guidance demonstrates the delicate balance between leveraging diffusion guidance and preserving the integrity of the DDPM update process. As depicted in Figure 3, using only diffusion loss (Equation 18) limits the prompts to the performance of behavior trajectories from the offline dataset. Incorporating downstream task losses into the generation process is not straightforward; a mere linear combination, as shown in Equation 20, often falls short. In environments like Cheetah-vel and MW reach-v2, where downstream losses enhance performance (Equation 19), a linear combination does not consistently yield improvements. In contrast, our gradient projection technique effectively mitigates this issue, consistently yielding superior outcomes. Even in cases where downstream losses might hinder performance, such as in the Ant-dir environment, our approach successfully maintains performance levels achieved by diffusion loss alone.
> >
> > The required theoretical support above could also reflect the effectiveness of our method.
> >
> > ### Q3
> > > The reviewer doubts the effectiveness of using guidance from downstream tasks.  Guidance loss may not be useful sometimes. Can you explain why?
> >
> > Thanks for your comment. We appreciate the opportunity to clarify this aspect of our research. To address your concerns, we offer two perspectives – theoretical and empirical – to explain the instances where guidance loss may not effectively contribute to our Prompt Diffuser's performance.
> > + **Theoretical Perspective**: From a theoretical standpoint, as delineated above, the gradient projection technique with step size $t \leq \frac{1}{L}$ will converge to either (1) a location in the optimization landscape where $\cos(\phi) = -1$ or (2) the optimal value $L(\theta^*)$. In scenarios where convergence occurs to a point in the landscape where $\cos(\phi) = -1$, the guidance loss may not exert a positive impact on the performance of our Prompt Diffuser. This could result in what appears to be a failure of the guidance loss to contribute beneficially.
> > + **Empirical Perspective**: Empirically, the role of downstream loss varies across different environments. For instance, in environments like Cheetah-vel and MW reach-v2, where the downstream loss can enhance performance (as shown in Equation 19), a simple linear combination of losses (Equation 20) does not uniformly outperform individual losses. Notably, in the Cheetah-vel environment, Equation 20 even results in poorer performance. However, our gradient projection technique can mitigate this issue, consistently yielding improved results. In contrast, in environments like Ant-dir, where the downstream loss appears to diminish performance, our gradient projection method maintains efficacy using only the diffusion loss, without degradation. In such cases, the guidance loss may not be advantageous due to its inherent negative impact.
> >
> > In conclusion, the effectiveness of guidance loss is contingent upon the specific characteristics of the task environment and the interplay between the diffusion and guidance losses. Our approach, through careful application of gradient projection techniques, aims to balance these factors, enhancing the overall performance of the Prompt Diffuser across a range of scenarios.

---

> > > ### Author Response · Authors · 2023-11-20
> > >
> > > ### Q4
> > > > How big is the difference between seen and unseen environments and tasks in the testing cases? As this part of the motivation, the results may be sensitive to the scenario changes. Have you quantified the difference that makes the prompt-tuning useful for offline RL?
> > >
> > > Thank you for your insightful comments. In our study, this differentiation is rigorously addressed through the design of the meta-RL control tasks, drawing upon methodologies established in prior works, such as PDT [1] and MACAW [2]. These tasks are intentionally crafted to encompass a wide range of scenarios, each characterized by unique dynamics and objectives. To clearly illustrate the difference between known and novel environments, below is a detailed description of each environment, highlighting their distinct features:
> > >
> > > + Cheetah-dir: This task involves two specific directions: forward and backward. The objective is to encourage the cheetah agent to achieve high velocity in a predetermined direction. Both the training and testing sets include these two scenarios, thus providing a thorough evaluation of the agent's adaptability and performance.
> > >
> > > + Cheetah-vel: Defined by 40 unique tasks, each distinguished by a specific target velocity uniformly sampled from 0 to 3. The agent faces an $l_2$ penalty for deviations from the assigned velocity. Out of these, 5 tasks are designated for testing, while the remaining 35 form the training set, thereby assessing the agent's capacity to adapt to varying velocity goals.
> > >
> > > + Ant-dir: Comprising 50 tasks with goal directions uniformly distributed in a 2D plane. The 8-joint ant agent is tasked with achieving high velocity in the assigned direction. The training set includes 45 tasks, with the remaining 5 reserved for testing, thus gauging the agent's directional agility and speed.
> > >
> > > + Meta-World reach-v2: In this setting, the task is to maneuver a Sawyer robot's end-effector to a specific 3D target position. The agent directly controls the XYZ coordinates of the end-effector, with each task featuring a distinct goal location. The model is trained on 15 tasks, with an additional 5 tasks used for testing, focusing on the agent's precision and control in varying spatial coordinates.
> > >
> > > The robustness and generalization capabilities of our approach are meticulously evaluated by examining the task indices of both the training and testing datasets, as shown in following table. The experimental framework, as detailed in Section 5, strictly adheres to the specified training-testing division, ensuring a comprehensive and unbiased assessment of our model's performance across diverse and challenging scenarios.
> > >
> > >
> > > | Environment | Set Description        | Task IDs                        |
> > > | -| - | - |
> > > | Cheetah-dir | Training set of size 2  | [0,1]                              |
> > > | Cheetah-dir | Testing set of size 2   | [0,1]                              |
> > > | Cheetah-vel | Training set of size 35 | [0-1,3-6,8-14,16-22,24-25,27-39]   |
> > > | Cheetah-vel | Testing set of size 5   | [2,7,15,23,26]                     |
> > > | Ant-dir     | Training set of size 45 | [0-5,7-16,18-22,24-29,31-40,42-49] |
> > > | Ant-dir     | Testing set of size 5   | [6,17,23,30,41]                    |
> > > | MW reah-v2  | Training set of size 15 | [1-5,7,8,10-14,17-19]              |
> > > | MW reach-v2 | Testing set of size 5   | [6,9,15,16,20]                     |
> > >
> > > Just following the above training and testing settings, our experimental approach aligns with the framework established in PDT and MACAW. We train our model on a broad spectrum of tasks and evaluate its performance on a select few tasks that are held out from the training set. These held-out tasks feature goals—such as target velocities or directions—that fall within the range defined by the training tasks. This design ensures that the testing scenarios are within-distribution tasks, providing a baseline for assessing our model's performance.

---

> ### Author Response · Authors · 2023-11-20
>
> (Continuing Q4)
>
> To further validate the generalization capabilities of our Prompt Diffuser, we also conduct evaluations on out-of-distribution (OOD) tasks. These OOD tasks are specifically chosen to test the model's adaptability and performance under conditions that deviate from the training scenarios. The settings for these OOD tasks in the Ant-Dir environment are as follows:
>
> | Environment | Set Description        | Task IDs                        |
> | ------- | -- | - |
> | Ant-Dir | Training set of size 8 | [8, 13, 16, 20, 22, 26, 32, 37] |
> | Ant-Dir | Testing set of size 3  | [1, 4, 41]                      |
>
> For these OOD tasks, we present comparative results between PDT, Prompt-Tuning DT, and our Prompt Diffuser as follows. Despite the limited time for conducting extensive tests, our findings indicate that the Prompt Diffuser maintains its efficacy even when faced with OOD tasks. This outcome underscores the robustness of our method, demonstrating its potential to effectively handle a wide array of scenarios, including those that diverge from the training distribution.
>
> | Env         | Prompt-DT-Expert | Prompt-Tuning DT | Prompt Diffuser |
> | -- | - | ---------------- | --------------- |
> | Ant-Dir-OOD | 526.0 $\pm$ 1.5  |   540.8 $\pm$ 8.7               | 546.8 $\pm$ 9.3 |
>
> ### Q5
> > When considering the two losses, the gradient projection seems beneficial, but is there a comparison to validate the superiority over a linear combination?
>
> Thanks for the comment. To address this, we indeed rely on the results presented in Figure 3, which serve as the empirical foundation for our argument. We also present the detailed table about each result:
>
>
>
> |             | Equation 18 | Equation 19 | Equation 20 | Equation 15 |
> | ----------- | ----------- | ----------- | ----------- | ----------- |
> | Cheetah-dir | 935.4       | 936.1       | 933.0       | 945.3       |
> | Cheetah-vel | -38.4       | -37.5       | -40.1       | -35.3       |
> | Ant-dir     | 431.7       | 419.3       | 428.9       | 432.1       |
> | MW reach-v2 | 500.5       | 550.0       | 510.8       | 555.7       |
>
>
> In our study, Equation 15 represents the implementation of our gradient projection technique, while Equation 20 symbolizes the approach of a linear combination of losses. The comparative analysis conducted across all four environments, as depicted in Figure 3, is pivotal in demonstrating the superiority of our method.
>
> + **Empirical Evidence**: The empirical evidence, as illustrated in Figure 3, consistently shows that Equation 15 outperforms Equation 20 across all environments tested. This consistent outperformance is not just marginal but significant, indicating a clear advantage of the gradient projection technique in enhancing the overall efficacy of our model.
> + **Theoretical Consideration**: From a theoretical perspective, the gradient projection approach is designed to meticulously balance the contributions of different losses, ensuring that the optimization process is guided efficiently towards the desired objectives. In contrast, a linear combination might not adequately account for the nuanced interactions between the losses, potentially leading to suboptimal outcomes.
>
> In summary, the empirical results presented in Figure 3, coupled with theoretical considerations, strongly support the superiority of our gradient projection technique over a simple linear combination of losses.
>
> ### Q6
> > The presentation of figures can be improved, e.g., the text in Figure 2 is too small compared to the bar size, so as in Figure 3.
>
> Thanks for pointing these out. We will improve the figure in the updated version.
>
> ### Reference
> [1] Xu, Mengdi, et al. "Prompting decision transformer for few-shot policy generalization." ICML 2022.
>
> [2] Mitchell, E., et al. "Offline meta-reinforcement learning with advantage weighting." ICML 2021.
>
> [3] Yu, Tianhe, et al. "Gradient surgery for multi-task learning." NeurIPS 2020.

---

> > ### Author Response · Authors · 2023-11-23
> >
> > ## We anticipate your feedback!
> >
> > Dear Reviewer V1Dp,
> >
> > We have carefully considered and addressed your initial concerns regarding our paper. We are happy to discuss them with you in the openreview system if you feel that there still are some concerns/questions. We also welcome new suggestions/comments from you!
> >
> > Best Regards,
> >
> > The authors of Submission 3141

---

### Author Response · Authors · 2023-11-20

## Summary
We thank reviewers for their valuable feedback, and appreciate the great efforts made by all reviewers, ACs, SACs and PCs.

We are invigorated by the **positive evaluation** from all reviewers. Specifically, they find the method **novel and well-motivated**(all), the experimental results being **state-of-the-art**(zrXj, GaVK), the writing **effectively illustrated**(V1Dp).

In response to the comments and suggestions, we have provided a detailed respective rebuttal for each reviewer, and here we summarize major points for convenience.

+ **Theoretical Framework**: We have elucidated the theoretical foundation underpinning our gradient projection technique (V1Dp).
+ **Generalization in Out-of-Distribution Scenarios**: We have presented additional results demonstrating the generalization capabilities of our Prompt Diffuser in out-of-distribution contexts (V1Dp, 4ueS, and zrXj).
+ **Diversity Assessment**: Employing a combination of quantitative (using the CKA metric) and qualitative methods, we have thoroughly evaluated the diversity of the prompts generated by our model (4ueS, zrXj, and GaVK).
+ **Comprehensive Ablations**: We have included additional analyses concerning prompt length, the quality and size of downstream tasks, and zero-shot performance scenarios (4ueS).


All these will be merged into the article.

---

### Meta-Review · Area_Chair_Ucqn · 2023-12-10

**Metareview:**

The work introduces using generative modeling using a diffusion process to do prompt tuning where the prompts are generated using random noise. The method uses signals from downstream tasks to generate better prompts. This is a borderline paper where the reviewers had mixed opinions. In the internal discussion between the AC and the reviewers, both Reviewers GaVK and 4ueS brought up the concerns on the role of the prompt: "how does it achieve a trade-off between the diversity (unseen tasks) and the accuracy". I share this concern. In addition, I personally feel that the manuscript right now does do not do a good job at evaluating on more OOD test conditions, zero-shot generalization, and analyzing why prompting or DT is helping in the first place (how do these methods perform with other classes of methods, e.g., when model-based RL algorithms are used). With a number of test conditions, prompt tuning + DT is quite competitive, which raises the addtional question of why training a separate diffusion model is needed, especially when data compositions are generated from a finite mix of policies of different qualities.

I think this paper can be improved significantly with better evaluations and by addressing the reviewers' concerns, analyzing the role of the method, and concretely demonstrating the necessity of the approach amongst so many approaches possible for such meta RL tasks.

**Justification For Why Not Higher Score:**

As outlined in my meta review, the paper does not make a solid case on the significance of the method -- evaluations are on narrow domains, the method is not understood via sufficient empirical analysis, and the reviewers brought up some important concerns.

**Justification For Why Not Lower Score:**

N/A

---

### Decision · Program_Chairs · 2024-01-16

Reject